# Group-in-Group Policy Optimization for LLM Agent Training

**Lang Feng**[1]    **Zhenghai Xue**[1]    **Tingcong Liu**[1]    **Bo An**[1,2,*]

[1]Nanyang Technological University, Singapore
[2]Skywork AI, Singapore
{lang005,zhenghai001,tingcong001}@e.ntu.edu.sg, boan@ntu.edu.sg

## Abstract

Recent advances in group-based reinforcement learning (RL) have driven frontier large language models (LLMs) in single-turn tasks like mathematical reasoning. However, their scalability to multi-turn LLM agent training remains limited. Unlike static tasks, agent-environment interactions unfold over many steps and often yield sparse or delayed rewards, making credit assignment across individual steps significantly more challenging. In this work, we propose *Group-in-Group Policy Optimization (GiGPO)*, a novel RL algorithm that achieves fine-grained credit assignment for LLM agents while preserving the appealing properties of group-based RL: critic-free, low memory, and stable convergence. GiGPO introduces a two-level structure for estimating relative advantage: (i) At the *episode-level*, GiGPO computes macro relative advantages based on groups of complete trajectories; (ii) At the *step-level*, GiGPO introduces an *anchor state grouping* mechanism that retroactively constructs step-level groups by identifying repeated environment states across trajectories. Actions stemming from the same state are grouped together, enabling micro relative advantage estimation. This hierarchical structure effectively captures both global trajectory quality and local step effectiveness without relying on auxiliary models or additional rollouts. We evaluate GiGPO on challenging agent benchmarks, including ALFWorld and WebShop, as well as tool-integrated reasoning on search-augmented QA tasks, using Qwen2.5-1.5B/3B/7B-Instruct. Crucially, GiGPO delivers fine-grained per-step credit signals, achieves performance gains of > 12% on ALFWorld and > 9% on WebShop over GRPO, and obtains superior performance on QA tasks (42.1% on 3B and 47.2% on 7B): all while maintaining the same GPU memory overhead, identical LLM rollout, and incurring little to no additional time cost.

## 1   Introduction

Large Language Models (LLMs) [1, 2, 3, 4] have leapt from static question-answer systems to versatile *agents* that perceive, reason, and act in open-ended environments. For instance, they now power embodied assistants that navigate simulated homes [5, 6], mobile and web navigators that plan multi-step browsing sessions [7, 8, 9, 10], and autonomous explorers in interactive games [11, 12]. In these settings, LLM agents need to perceive, reason, and act in multi-turn loops, which requires not only language understanding but also long-horizon planning and decision-making.

Reinforcement learning (RL) [13] has become a crucial recipe for post-training LLMs, leading to frontier models like OpenAI o1 [14] and DeepSeek R1 [15]. In particular, group-based RL algorithms such as RLOO [16, 17] and GRPO [18] have proven especially effective in large-scale training. These

---

*Corresponding author
Code: https://github.com/langfengQ/verl-agent

methods replace value-function estimation with simple yet powerful relative advantage estimation within groups of rollouts. This group-based advantage computation enjoys favorable properties such as low memory overhead, critic-free optimization, and scalability to large models. However, their successes have so far been largely limited to single-turn tasks such as math problem solving [19, 20] and code generation [21], where reward arrives immediately and credit assignment is straightforward.

In contrast, LLM agents operating in external environments face fundamentally different learning landscapes. Their behavior unfolds over long episodes with tens of decision steps and tens of thousands of tokens (e.g., an ALFWorld [5] episode may include up to 50 steps and over 20k tokens). Rewards are typically sparse (sometimes arriving only at the end of an episode), and the impact of any individual action may only manifest much later in the trajectory. This substantially complicates the credit assignment for individual steps and increases the challenge of policy optimization. Naively applying existing group-based RL algorithms in such settings collapses step-level distinctions, undermining their effectiveness. Hence, these limitations raise a core question:

> *Can we preserve the critic-free, low-memory, and stable convergence properties of group-based RL while introducing fine-grained credit assignment for multi-turn LLM agent training?*

To address this, we introduce *Group-in-Group Policy Optimization* (GiGPO), a new group-based RL algorithm that nests two-dimensional notions of credit assignment, better-suited for the multi-turn optimization of LLM agents. (i) At the *episode level*, GiGPO samples a group of complete trajectories under identical task and initial-state conditions, and computes macro relative advantages based on total returns like vanilla GRPO [18]. This captures the overall effectiveness of each trajectory and reflects the completeness of task execution. (ii) At the *step level*, GiGPO introduces a novel *anchor state grouping* mechanism for fine-grained relative advantages estimation. Specifically, it retroactively identifies the repeated environment states, termed as *anchor states*, across trajectories and uses them as anchors to construct step-level groups that allow for localized credit assignment.

The key insight behind GiGPO is that, under identical tasks and initial environment conditions, many trajectories within the group encounter the same states *multiple times* due to ineffective actions or loops, such as revisiting the same webpage, room, or game scene. These shared states provide a natural basis for step-level group construction and computing more granular advantage estimates. GiGPO uses these step-level groups to assign localized credit to actions based on their relative performance at a common state, enabling more precise optimization while avoiding the cost explosion of per-step extra rollouts. As such, GiGPO remains fully critic-free and requires no auxiliary value models while dramatically introducing finer credit signals for training LLM agents.

We first evaluate GiGPO on long-horizon agent benchmarks: ALFWorld [5], which tests embodied task planning in simulated household environments, and WebShop [22], which simulates complex, goal-driven web interactions. In addition, we study multi-turn tool-integrated reasoning on search-augmented QA tasks. Our experiments with Qwen2.5-1.5B/3B/7B-Instruct [3] show that GiGPO consistently outperforms prompt-based agents, actor-critic baselines, and prior group-based RL methods. In particular, GiGPO injects fine-grained, step-level credit signals that sharpen policy learning of agents over horizons and achieves performance gains of > 12% on ALFWorld and > 9% on WebShop over GRPO, along with remarkable performance on search-based QA tasks (42.1% on 3B and 47.2% on 7B). These gains come without compromising the core strengths of group-based RL (only < 0.002% time cost), making GiGPO a versatile and high-utility algorithm for LLM agents.

## 2 Related Work

**LLMs as decision-making agents.** The use of large language models (LLMs) as autonomous agents has expanded rapidly across domains such as program generation [23], smart device operation [24, 25, 26, 27], interactive gameplay [11], and robot behavior control [28]. Early works typically relied on leveraging pre-trained, frozen models through carefully designed prompting methods (like ReAct [29] and Reflexion [30]), enhanced memory and retrieval systems [12, 31], and integration with external tools [32, 33, 34]. More recent research has shifted toward adapting model parameters with supervised fine-tuning (SFT) [24] or RL [13], enabling agents to learn directly from environment interaction rather than static prompts or handcrafted workflows, which we introduce below.

**Reinforcement learning for LLM agents.** RL has played a pivotal role in enabling LLM agents to operate in dynamic, open-ended environments. Early work applied classical RL algorithms such as

DQN [35] to train LLM agents in text-based games [36] and later research [37, 38, 39, 40, 41] started to employ value-based methods, such PPO [42] and AWR [43], in more diverse and interactive agent scenarios including Android device control [44], embodied ALFWorld [5], and card games [45]. More recent approaches have extended RL training to complex web-based and application-centered tasks. For instance, ArCHer [46] and AgentQ [47] target the WebShop benchmark [22], but require intricate designs and computation overhead such as additional value networks or Monte Carlo Tree Search (MCTS) [48]. CoSo [10] introduces an entropy-based RL method that enhances the performance of agents. Going further, LOOP [49] introduces a hybrid method combining REINFORCE leave-one-out (RLOO) [16, 17] with PPO-style updates, achieving state-of-the-art results in AppWorld [50]. RAGEN [51] introduces a trajectory-level GRPO that concatenates all states, intermediate reasoning, and actions into a unified episode-level response. However, it faces scalability challenges in long-horizon tasks (e.g., in ALFWorld, which involves up to 50 steps).

**Reinforcement learning for large language models.** An early and influential application of RL in LLMs is the Reinforcement Learning from Human Feedback (RLHF) [52, 53, 54, 55], which focuses on aligning LLMs to human preferences. Most recent works have explored using RL to enhance the capabilities of reasoning and logical deduction in LLMs [56, 15]. In particular, group-based RL algorithms have emerged as promising alternatives to traditional methods like PPO [42]. These methods, such as RLOO [16, 17], GRPO [18], Dr. GRPO [19], DAPO [20], and CPPO [57], avoid introducing extra value functions by leveraging a group of samples from the same query and estimate the advantages accordingly. This enables the large-scale RL training and has shown strong results in tasks such as mathematical reasoning [15], search [58, 59], and tool use [60, 61]. Our work is closely related to this line of research, with a focus on training *LLM agents*. We aim to retain the benefits of group-based RL, such as critic-free learning and efficiency, while introducing finer-grained credit assignment. Moreover, the hierarchical core of GiGPO is orthogonal to existing group-based RL approaches, making it fully compatible and capable of incorporating them to enhance performance.

## 3 Preliminaries

**Problem setup.** We consider a general setting in which an LLM agent interacts with an environment to complete multi-step tasks based on a task description $x \in p(X)$. At each discrete time step $t = 1, 2, \ldots, T$, the agent observes a state $s_t \in \mathcal{S}$ and generates a textual action $a_t \in \mathcal{V}^n$, where $\mathcal{V}$ denotes the token vocabulary and $n$ is the maximum generation length. The environment then returns a scalar reward $r_t \in \mathbb{R}$ and the next state $s_{t+1}$. A full episode consists of a trajectory $\boldsymbol{\tau} = \{(s_1, a_1, r_1), (s_2, a_2, r_2), \ldots, (s_T, a_T, r_T)\}$. The agent's behavior is governed by an LLM policy $\pi_\theta(a_t | s_t, x)$, parameterized by $\theta$, which defines a distribution over outputs conditioned on the current state $s_t$ and the task prompt $x$. In many realistic scenarios, the environment may provide sparse or delayed rewards (e.g., success and failure signals at the end of an episode) or weak feedback signals for intermediate steps. As the agent generates $T$ consecutive textual actions $(a_1, \ldots, a_T)$, each potentially spanning thousands of tokens, it becomes particularly challenging to assign credit to individual tokens over the course of an episode.

**Group-based RL.** Recent RL works converge on a simple recipe for training LLMs: for a given task description $x$, the LLM samples a group of $N$ candidate trajectories $\{\boldsymbol{\tau}_1, \boldsymbol{\tau}_2, \ldots, \boldsymbol{\tau}_N\}$, each corresponding to one full episode rollout under $\pi_{\theta_{\text{old}}}$. Each trajectory $\boldsymbol{\tau}_i$ receives a scalar reward $R(\boldsymbol{\tau}_i)$ reflecting the overall quality or success of the generated outcome. Instead of estimating advantages using separate value functions like PPO [42], group-based RL methods compute advantages purely based on the statistics internal to the sampled group:

$$A(\boldsymbol{\tau}_i) = \texttt{GroupComputation}(\{R(\boldsymbol{\tau}_i)\}_{i=1}^N). \tag{1}$$

For example, in GRPO [18], the advantage of each trajectory is estimated by normalizing its reward with respect to the group's mean and standard deviation. This design is highly memory-efficient and can scale effectively to the large batch sizes and model sizes typical in modern LLM training, making it a practical and scalable choice for large-scale RL training.

## 4 Training LLM Agents with GiGPO

While group-based RL algorithms [18, 15] have proven highly effective for training LLMs in single-turn tasks, their extension to multi-step agent settings faces critical challenges in credit assignment.

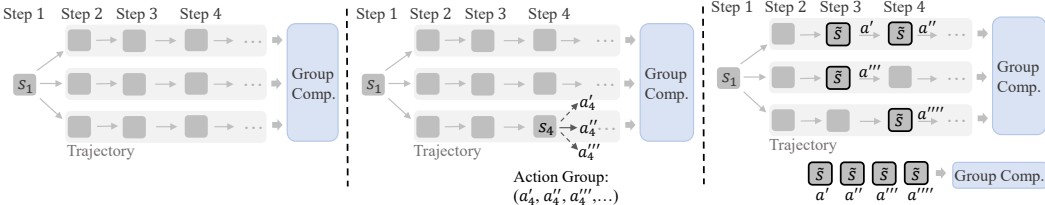

Figure 1: Comparison of multi-turn LLM agent training. **Left**: Vanilla GRPO rolls out a group of full trajectories and computes episode-level advantages. **Middle**: Constructing step-level groups via additional per-state rollouts (e.g., $a'_4, a''_4, a'''_4, \dots$) enables fine-grained feedback but incurs prohibitive computational cost. **Right**: GiGPO efficiently achieves fine-grained credit assignment by aggregating distinct actions ($a', a'', a''', a''''$) taken from the same environment state $\tilde{s}$ across the trajectories.

Figure 1 illustrates this gap. Vanilla GRPO (left) treats each trajectory as a whole and computes a single relative advantage for the entire episode, which fails to provide actionable feedback for individual steps. A natural remedy is to roll out multiple single-step actions for each state $s_t$ via $\pi_{\theta_{\text{old}}}$ as shown in Figure 1 (middle). However, this approach quickly becomes impractical due to the substantial overhead of extra LLM forward passes and the difficulty of evaluating rewards for hypothetical actions never actually executed.

To overcome these challenges, we propose our *Group-in-Group Policy Optimization (GiGPO)* in this section. Similar to prior works [49, 51], GiGPO begins by sampling groups of trajectories under identical tasks and initial environment states. It then introduces a two-level grouping structure: preserving episode-level grouping for holistic performance comparison, while dynamically constructing an additional set of step-level groups by retroactively aggregating actions encountering the same environment states. This "group-in-group" construction yields two complementary advantages: (1) *Episode relative advantages* capture the holistic effectiveness of each trajectory, providing a stable, global training signal. (2) *Step relative advantages* zoom in on which actions outperform their peers within the same state, endowing the gradient with fine-grained credit.

Figure 2 presents an overview of the GiGPO training pipeline. In the remainder of this section, we will detail the computation of episode-level relative advantages, elaborate on the anchor state grouping mechanism, describe the derivation of step-level relative advantages, and finally present the overall GiGPO objective.

## 4.1 Episode Relative Advantages

We first introduce the episode-level relative advantages, which represent the coarse-grained component of GiGPO and mirror the naive application of GRPO at the trajectory level. We roll out the agent's policy $\pi_{\theta_{\text{old}}}$ in the environment to collect $N$ complete trajectories under a fixed task $x$ and identical initial states. Formally, this process yields a group of trajectories $\{\tau_i\}_{i=1}^{N}$, where each trajectory is denoted as $\tau_i = \{(s_1^{(i)}, a_1^{(i)}, r_1^{(i)}), \dots, (s_T^{(i)}, a_T^{(i)}, r_T^{(i)})\}$ and the initial states satisfy $s_1^{(1)} = s_1^{(2)} = , \dots, = s_1^{(N)}$. For each trajectory, we utilize the total return $R(\tau_i) = \sum_t r_t^{(i)}$ as a holistic measure of how effectively the agent completes the task. In settings where only a binary reward is given at the end of the episode, this simplifies to $R(\tau_i) = 1$ for success and $R(\tau_i) = 0$ for failure. Then, we organize the trajectories and their corresponding returns into an episode-level group:

$$G^E = \left\{ \big(\tau_1, R(\tau_1)\big), \big(\tau_2, R(\tau_2)\big), \dots, \big(\tau_N, R(\tau_N)\big) \right\}. \tag{2}$$

To evaluate the global relative quality of each trajectory within the group, we compute an episode relative advantage $A^E(\tau_i)$ for each $\tau_i$ by normalizing the total return with the group's mean and a normalization factor:

$$A^E(\tau_i) = \frac{R(\tau_i) - \text{mean}\big(\{R(\tau_j)\}_{j=1}^{N}\big)}{F_{\text{norm}}\big(\{R(\tau_j)\}_{j=1}^{N}\big)}. \tag{3}$$

In GRPO [18], the default normalization factor is defined as the standard deviation, i.e., $F_{\text{norm}} = \text{std}$. However, this may introduce a difficulty bias [19], where trajectories from low-variance groups (e.g., very easy or hard tasks) receive disproportionately large gradients. In the context of the LLM

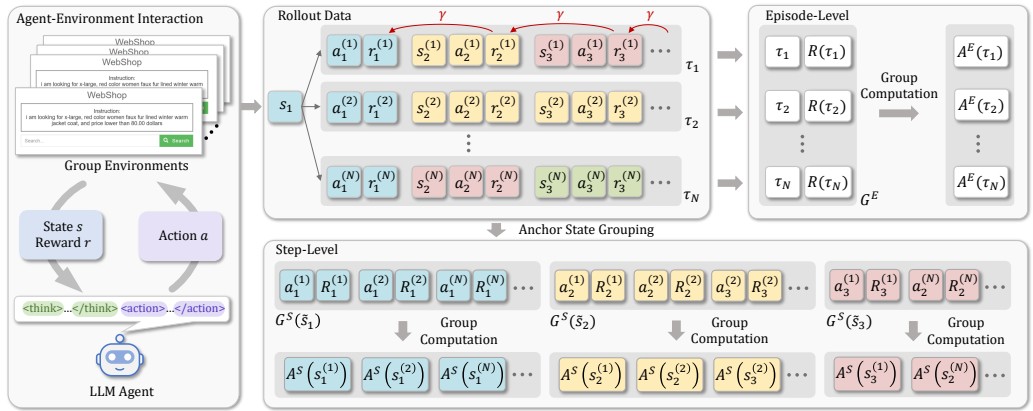

Figure 2: Overview of GiGPO. The agent interacts with a group of environments initialized with identical states to generate a set of trajectories $\{\boldsymbol{\tau}_i\}_{i=1}^N$. States with the same color represent the same environment state. GiGPO performs two-dimensional group computations (episode-level $A^E$ and step-level $A^S$) to produce hierarchical relative advantages that guide fine-grained policy optimization.

agent, where tasks often involve very long horizons, this effect tends to emerge frequently, potentially affecting the stability of updates. As an alternative, we also consider a fixed normalization factor $F_{\text{norm}} = 1$, which yields an unbiased Leave-One-Out estimator [16] (see Appendix C for details). This simple adjustment helps stabilize training in some challenging agent scenarios.

Overall, the episode relative advantage captures whether the agent successfully completes the assignment across the entire decision horizon $T$. Similar to the vanilla GRPO for multi-step optimization shown in Figure 1 (left), it primarily focuses on macro credit assignment, encouraging the policy to develop coherent, trajectory-wide behaviors that maximize overall task performance.

### 4.2 Step Relative Advantages

While the episode relative advantage offers a macro, trajectory-wide signal, it cannot distinguish between the contributions of individual actions within the trajectory. To obtain this fine-grained feedback, we need to form step-level groups: for the same state, we gather the different actions and compare their outcomes, thereby learning which choices are relatively better or worse. A naive way to do so would be to roll out fresh actions from every state (Figure 1, middle), but that is prohibitively expensive. Instead, we introduce anchor state grouping below, avoiding extra LLM overhead.

**Anchor state grouping.** As all trajectories $\{\boldsymbol{\tau}_1, \ldots, \boldsymbol{\tau}_N\}$ arise from the same task $x$ and identical initial conditions, many environment states naturally recur across episodes and even across time steps within a single trajectory. We leverage this redundancy by identifying and grouping identical states across trajectories, thereby dynamically constructing step-level groups. Specifically, let $\mathcal{U} = \{\tilde{s}_1, \tilde{s}_2, \ldots, \tilde{s}_U\}$ denote the set of all distinct environment states appearing in the trajectory group $\{\boldsymbol{\tau}_1, \ldots, \boldsymbol{\tau}_N\}$. We treat each such unique state $\tilde{s} \in \mathcal{U}$ as an implicit anchor and use it to gather all matching occurrences of that state, and therefore call $\tilde{s}$ as "*anchor state*". Based on this, we can construct $|\mathcal{U}|$ step-level groups (one for each unique anchor state $\tilde{s}$), which is defined as follows:

$$G^S(\tilde{\boldsymbol{s}}) = \left\{ \left(\boldsymbol{a}_t^{(i)}, r_t^{(i)}\right) \;\middle|\; \boldsymbol{s}_t^{(i)} = \tilde{\boldsymbol{s}},\; 1 \le i \le N,\; 1 \le t \le T \right\}. \tag{4}$$

Unlike per-state rollout, this procedure incurs no extra rollouts: it is entirely offline and requires only lightweight key-based grouping using hashmaps. Each group $G^S(\tilde{\boldsymbol{s}})$ contains multiple instances of the same environment state paired with potentially different actions. Hence, this structure effectively builds the step-level group, forming the basis for subsequent step-level advantage estimation.

**Relative advantage computation.** Although each tuple $\left(\boldsymbol{a}_t^{(i)}, r_t^{(i)}\right)$ contains an immediate reward $r_t^{(i)}$, it may be sparse, especially in long-horizon tasks. To better capture long-term impact, we associate a *discounted return* with each step. Let $\gamma \in (0, 1]$ be the standard RL discount factor. For each element in $G^S(\tilde{\boldsymbol{s}})$, we compute its discounted return $R_t^{(i)}$ by

$$R_t^{(i)} = \sum\nolimits_{k=t}^{T} \gamma^{k-t} r_k^{(i)}. \tag{5}$$

This quantity captures the future impact of action $\boldsymbol{a}_t^{(i)}$ on subsequent rewards, rather than relying solely on the immediate reward $r_t^{(i)}$. Accordingly, the step-level group for each $\tilde{\boldsymbol{s}} \in \mathcal{U}$ becomes:

$$G^S(\tilde{\boldsymbol{s}}) = \left\{ \left(\boldsymbol{a}_t^{(i)}, R_t^{(i)}\right) \mid \boldsymbol{s}_t^{(i)} = \tilde{\boldsymbol{s}}, \ 1 \leq i \leq N, \ 1 \leq t \leq T \right\}. \tag{6}$$

Once these step-level groups are formed, we compute the *step relative advantage* for each $\tilde{\boldsymbol{s}} \sim \mathcal{U}$ and each action $\boldsymbol{a}_t^{(i)}$ in $G^S(\tilde{\boldsymbol{s}})$:

$$A^S(\boldsymbol{a}_t^{(i)}) = \frac{R_t^{(i)} - \text{mean}\left(\left\{R_t^{(j)} \mid (\boldsymbol{a}_t^{(j)}, R_t^{(j)}) \in G^S(\tilde{\boldsymbol{s}})\right\}\right)}{F_{\text{norm}}\left(\left\{R_t^{(j)} \mid (\boldsymbol{a}_t^{(j)}, R_t^{(j)}) \in G^S(\tilde{\boldsymbol{s}})\right\}\right)}. \tag{7}$$

$A^S$ provides micro credit assignment and fine-grained feedback on the relative quality of individual actions taken from the same state. In contrast to the coarse, trajectory-wide signal of $A^E$, it offers step-level guidance that is essential for refining decisions in long-horizon agent tasks.

**How does step-level group work?** We present an intuitive illustration in Figure 3 to show the utility of the step relative advantages. We consider two example trajectories from the set $\{\boldsymbol{\tau}_i\}_{i=1}^N$. In $\tau_1$, the agent first selects the *2nd Item* (incorrect), then returns to the previous page and selects the *1st Item* (correct), successfully completing the task. Due to temporal discounting (Equation (5)), the earlier action (*2nd Item*) receives a lower discounted return than the later correct one (*1st Item*). In $\tau_2$, the agent clicks the *Next Page*, ultimately failing to find the target and receiving no reward. By aggregating these actions into the same step-level group based on their shared anchor state, GiGPO computes their relative advantages and yields a clear preference ordering: $A^S(\textit{1st Item}) > A^S(\textit{2nd Item}) > A^S(\textit{Next Page})$. This ranking successfully captures fine-grained distinctions in long-term utility that are missed by prior group-based RL methods [17, 18, 20]. While this example illustrates GiGPO's effectiveness in sparse-reward environments, its advantages extend naturally to dense-reward scenarios, where per-step rewards can be fully leveraged to assess the relative quality of individual actions within shared states.

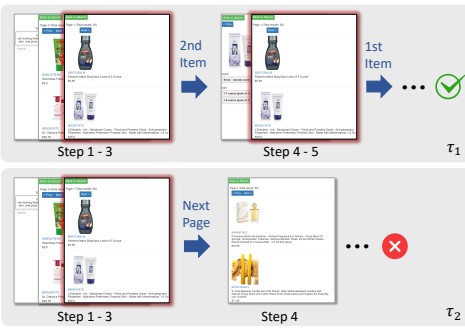

Figure 3: Illustration of step-level grouping in WebShop. Both $\tau_1$ and $\tau_2$ encounter the same environment state multiple times: a search results page (highlighted by the red border). **Top**: $\tau_1$ eventually succeeds. **Bottom**: $\tau_2$ leads to failure.

### 4.3 Group-in-Group Policy Optimization

We finally combine the two levels of advantage signals into a single *group-in-group advantage* to assign credit at both global (episode) and local (step) scales:

$$A(\boldsymbol{a}_t^{(i)}) = A^E(\boldsymbol{\tau}_i) + \omega \cdot A^S(\boldsymbol{a}_t^{(i)}), \tag{8}$$

where $\omega \in \mathbb{R}_{\geq 0}$ is a weighting coefficient that balances episode relative advantage and step relative advantage. $A^E(\boldsymbol{\tau}_i)$ captures how good an episode is compared to others in the group, while $A^S(\boldsymbol{a}_t^{(i)})$ refines step-level performance within shared environment state conditions. Jointly, they provide hierarchical supervision for the policy optimization of LLM agents. Then the clipped policy optimization objective of GiGPO is:

$$\mathcal{J}_{\text{GiGPO}}(\theta) = \mathbb{E}_{\substack{x \sim p(X) \\ \{\boldsymbol{\tau}_i\}_{i=1}^N \sim \pi_{\theta_{\text{old}}}}} \left[ \frac{1}{NT} \sum_{i=1}^N \sum_{t=1}^T \min\left(\rho_\theta(\boldsymbol{a}_t^{(i)}) A(\boldsymbol{a}_t^{(i)}), \text{clip}\left(\rho_\theta(\boldsymbol{a}_t^{(i)}), 1 \pm \epsilon\right) A(\boldsymbol{a}_t^{(i)})\right)\right]$$
$$- \beta \mathbb{D}_{\text{KL}}\left(\pi_\theta(\cdot \mid x) \| \pi_{\text{ref}}(\cdot \mid x)\right). \tag{9}$$

where $\rho_\theta(\boldsymbol{a}_t^{(i)}) = \frac{\pi_\theta(\boldsymbol{a}_t^{(i)} \mid \boldsymbol{s}_t^{(i)}, x)}{\pi_{\theta_{\text{old}}}(\boldsymbol{a}_t^{(i)} \mid \boldsymbol{s}_t^{(i)}, x)}$ is the importance sampling ratio, $\beta$ controls the strength of the KL penalty encouraging proximity to a reference policy $\pi_{\text{ref}}$. We present the pseudo code in Appendix D.

# 5 Experiment

In this section, we present empirical evaluations of GiGPO across a variety of agentic tasks. Specifically, we aim to demonstrate: **(1)** the strong ability of GiGPO in training LLM agents; **(2)** the ablation study of GiGPO; **(3)** the dynamic trend of step-level group $G^S(\tilde{s})$ over the course of training; **(4)** the computational budget of GiGPO.

## 5.1 Experiment Setup

**Benchmarks.** We first train the LLM agents on two challenging benchmarks: ALFWorld [5] and WebShop [22]. *ALFWorld* is an embodied environment designed to assess the ability of LLM agents to perform multi-step decision-making. In each episode, the agent receives a text goal and must accomplish it through multi-turn interaction with the environment. It includes 3,827 task instances across six categories of common household activities: Pick & Place (Pick), Examine in Light (Look), Clean & Place (Clean), Heat & Place (Heat), Cool & Place (Cool), and Pick Two & Place (Pick2). *WebShop* is a complex, web-based interactive environment designed to test the LLM agents in realistic online shopping scenarios. To complete the task, the agent must interact with a simulated HTML-based shopping website to search for, navigate to, and ultimately purchase a suitable item. It contains over 1.1 million products and 12k user instructions, providing a rich and diverse action space. In addition, we also evaluate the multi-turn tool calling performance of GiGPO on *search-augmented QA tasks*, including single-hop QA datasets (NQ [62], TriviaQA [63], and PopQA [64]) and multi-hop QA datasets (HotpotQA [65], 2Wiki [66], MuSiQue [67], and Bamboogle [68]).

**Baselines.** For ALFWorld and WebShop, we compare our approach with a range of competitive baselines: (1) Closed-source LLMs: GPT-4o [1] and Gemini-2.5-Pro [2], which represent state-of-the-art capabilities in general-purpose reasoning and language understanding. (2) Prompting agents: ReAct [29] and Reflexion [30], which rely on in-context prompting to guide multi-step behavior without parameter updates. (3) RL training methods: PPO [42], a widely used actor-critic algorithm that requires an additional value model, and group-based critic-free methods RLOO [16, 17] and GRPO [18], which perform advantage estimation over trajectory groups. For search-augmented QA, we compare GiGPO with R1-Instruct, Search-R1 [58], ZeroSearch [59], and StepSearch [69].

**Training details.** We use Qwen2.5-1.5B/3B/7B-Instruct [3] as our base models. The weighting coefficient $\omega$ is set to 1 with no further tuning. For ALFWorld and WebShop, all RL training methods (including ours and the baselines) use exactly the same hyperparameter configurations. The rollout group size $N$ for group-based RL methods is set to 8. For search-augmented QA, we follow the same settings in Search-R1 [58]. We use E5 [70] as the retriever. The rollout group size $N$ is set to 5 and the max turn is set to 4. Moreover, we incorporate similarity-based GiGPO, where anchor state grouping is performed by grouping two states if their similarity (longest matching subsequence) exceeds the threshold of 0.9. Full training settings and hyperparameter details are provided in Appendix E.1.

## 5.2 Performance on ALFWorld and WebShop

Table 1 demonstrates the strong performance of GiGPO across both ALFWorld and WebShop. As shown, closed-source LLMs offer only moderate performance: Gemini-2.5-Pro reaches 60.3% success on ALFWorld and 35.9% on WebShop, while GPT-4o lags further behind. Open-source prompt-only agents (e.g., ReAct and Reflexion) show marginal improvements over vanilla prompting but still underperform, underscoring the difficulty of long-horizon control without post-training. RL training brings substantial gains: PPO improves average ALFWorld success to 54.4% on the 1.5B model and 80.4% on the 7B model, with WebShop scores also increasing significantly. However, this comes at the expense of increased complexity: requiring a separate critic network, hyperparameter tuning, and longer training durations [71, 49]. GRPO and RLOO also yield strong performance while being more computationally efficient, demonstrating the effectiveness of group-based RL in large-scale LLM training. Nevertheless, their lack of fine-grained per-step feedback limits their ability to provide precise credit assignment across long horizons. In contrast, GiGPO overcomes this limitation with a two-level advantage estimation, enabling both GiGPO$_{\text{w/ std}}$ and GiGPO$_{\text{w/o std}}$ to consistently surpass GRPO and RLOO. In particular, GiGPO$_{\text{w/o std}}$ surpasses GRPO by 13.3% on ALFWorld and 10.6% on WebShop at 1.5B, and by 12.6% and 9.1%, respectively, at 7B. These results highlight GiGPO's superior ability to train LLM agents more effectively and more efficiently. We also find that GiGPO enables agents to exhibit emergent reasoning behavior (see Appendix F).

Table 1: Performance on ALFWorld and WebShop. Results are averaged over 3 random seeds. For ALFWorld, we report the average success rate (%) for each subtask as well as the overall result. For WebShop, we report both the average score and the average success rate (%). $\text{GiGPO}_{\text{w/ std}}$ denotes using $F_{\text{norm}} = \text{std}$, while $\text{GiGPO}_{\text{w/o std}}$ uses $F_{\text{norm}} = 1$.

| Type | Method | ALFWorld | | | | | | | WebShop | |
| --- | --- | --- | --- | --- | --- | --- | --- | --- | --- | --- |
| | | Pick | Look | Clean | Heat | Cool | Pick2 | All | Score | Succ. |
| *Closed-Source Model* | | | | | | | | | | |
| Prompting | GPT-4o | 75.3 | 60.8 | 31.2 | 56.7 | 21.6 | 49.8 | 48.0 | 31.8 | 23.7 |
| Prompting | Gemini-2.5-Pro | 92.8 | 63.3 | 62.1 | 69.0 | 26.6 | 58.7 | 60.3 | 42.5 | 35.9 |
| *Qwen2.5-1.5B-Instruct* | | | | | | | | | | |
| Prompting | Qwen2.5 | 5.9 | 5.5 | 3.3 | 9.7 | 4.2 | 0.0 | 4.1 | 23.1 | 5.2 |
| Prompting | ReAct | 17.4 | 20.5 | 15.7 | 6.2 | 7.7 | 2.0 | 12.8 | 40.1 | 11.3 |
| Prompting | Reflexion | 35.3 | 22.2 | 21.7 | 13.6 | 19.4 | 3.7 | 21.8 | 55.8 | 21.9 |
| RL Training | PPO (with critic) | $64.8_{\pm3.5}$ | $40.5_{\pm6.9}$ | $57.1_{\pm4.9}$ | $60.6_{\pm6.6}$ | $46.4_{\pm4.0}$ | $47.4_{\pm1.9}$ | $54.4_{\pm3.1}$ | $73.8_{\pm3.0}$ | $51.5_{\pm2.9}$ |
| RL Training | RLOO | $88.3_{\pm3.0}$ | $52.8_{\pm8.6}$ | $71.0_{\pm5.9}$ | $62.8_{\pm8.7}$ | $66.4_{\pm5.5}$ | $56.9_{\pm4.7}$ | $69.7_{\pm2.5}$ | $73.9_{\pm5.6}$ | $52.1_{\pm6.7}$ |
| RL Training | GRPO | $85.3_{\pm1.5}$ | $53.7_{\pm8.0}$ | $84.5_{\pm6.8}$ | $78.2_{\pm7.9}$ | $59.7_{\pm5.0}$ | $53.5_{\pm5.6}$ | $72.8_{\pm3.6}$ | $75.8_{\pm3.5}$ | $56.8_{\pm3.8}$ |
| RL Training | $\text{GiGPO}_{\text{w/ std}}$ | $94.4_{\pm5.9}$ | $67.5_{\pm4.6}$ | $94.8_{\pm3.8}$ | $94.4_{\pm7.8}$ | $79.8_{\pm4.7}$ | $76.4_{\pm5.4}$ | $86.7_{\pm1.7}$ | $83.1_{\pm1.6}$ | $65.0_{\pm3.2}$ |
| RL Training | $\text{GiGPO}_{\text{w/o std}}$ | $96.0_{\pm1.4}$ | $76.5_{\pm3.9}$ | $91.8_{\pm5.5}$ | $91.3_{\pm6.3}$ | $71.7_{\pm8.4}$ | $79.5_{\pm7.7}$ | $86.1_{\pm4.7}$ | $83.5_{\pm1.8}$ | $67.4_{\pm4.5}$ |
| *Qwen2.5-7B-Instruct* | | | | | | | | | | |
| Prompting | Qwen2.5 | 33.4 | 21.6 | 19.3 | 6.9 | 2.8 | 3.2 | 14.8 | 26.4 | 7.8 |
| Prompting | ReAct | 48.5 | 35.4 | 34.3 | 13.2 | 18.2 | 17.6 | 31.2 | 46.2 | 19.5 |
| Prompting | Reflexion | 62.0 | 41.6 | 44.9 | 30.9 | 36.3 | 23.8 | 42.7 | 58.1 | 28.8 |
| RL Training | PPO (with critic) | $92.3_{\pm4.0}$ | $64.0_{\pm8.4}$ | $92.5_{\pm2.4}$ | $89.5_{\pm7.0}$ | $80.3_{\pm2.0}$ | $68.8_{\pm8.3}$ | $80.4_{\pm2.7}$ | $81.4_{\pm3.1}$ | $68.7_{\pm5.1}$ |
| RL Training | RLOO | $87.6_{\pm4.3}$ | $78.2_{\pm8.3}$ | $87.3_{\pm5.8}$ | $81.3_{\pm7.6}$ | $71.9_{\pm5.2}$ | $48.9_{\pm8.4}$ | $75.5_{\pm4.6}$ | $80.3_{\pm3.2}$ | $65.7_{\pm4.0}$ |
| RL Training | GRPO | $90.8_{\pm5.1}$ | $66.1_{\pm6.7}$ | $89.3_{\pm5.4}$ | $74.7_{\pm6.9}$ | $72.5_{\pm5.4}$ | $64.7_{\pm7.3}$ | $77.6_{\pm5.2}$ | $79.3_{\pm2.8}$ | $66.1_{\pm3.7}$ |
| RL Training | $\text{GiGPO}_{\text{w/ std}}$ | $97.7_{\pm1.6}$ | $82.7_{\pm7.9}$ | $98.8_{\pm1.6}$ | $83.7_{\pm7.2}$ | $89.3_{\pm8.2}$ | $79.2_{\pm6.6}$ | $90.8_{\pm1.3}$ | $84.4_{\pm2.9}$ | $72.8_{\pm3.2}$ |
| RL Training | $\text{GiGPO}_{\text{w/o std}}$ | $91.8_{\pm5.4}$ | $88.6_{\pm6.3}$ | $95.9_{\pm3.2}$ | $90.2_{\pm2.6}$ | $86.5_{\pm5.5}$ | $85.2_{\pm7.5}$ | $90.2_{\pm2.3}$ | $86.2_{\pm2.6}$ | $75.2_{\pm3.8}$ |

Table 2: Performance on search-augmented QA tasks. GiGPO is trained on NQ and HotpotQA with $F_{\text{norm}} = \text{std}$. † and ⋆ indicate in-domain and out-of-domain datasets, respectively.

| Type | Method | Single-Hop QA | | | Multi-Hop QA | | | | Avg. |
| --- | --- | --- | --- | --- | --- | --- | --- | --- | --- |
| | | NQ† | TriviaQA⋆ | PopQA⋆ | HotpotQA† | 2Wiki⋆ | MuSiQue⋆ | Bamboogle⋆ | |
| *Qwen2.5-3B-Instruct* | | | | | | | | | |
| RL Training | R1-Instruct | 27.0 | 53.7 | 19.9 | 23.7 | 29.2 | 7.2 | 29.3 | 27.1 |
| RL Training | Search-R1 | 34.1 | 54.5 | 37.8 | 32.4 | 31.9 | 10.3 | 26.4 | 32.5 |
| RL Training | ZeroSearch | 41.4 | 57.4 | **44.8** | 27.4 | 30.0 | 9.8 | 11.1 | 31.7 |
| RL Training | StepSearch | - | - | - | 34.5 | 32.0 | **17.4** | 34.4 | - |
| RL Training | **GiGPO** | **42.0** | **59.5** | 42.4 | **36.9** | **37.0** | 12.6 | **64.1** | **42.1** |
| *Qwen2.5-7B-Instruct* | | | | | | | | | |
| RL Training | R1-Instruct | 21.0 | 44.9 | 17.1 | 20.8 | 27.5 | 6.0 | 19.2 | 22.4 |
| RL Training | Search-R1 | 39.3 | 61.0 | 39.7 | 37.0 | 40.1 | 14.6 | 36.8 | 38.5 |
| RL Training | ZeroSearch | 43.6 | 61.8 | **51.5** | 34.6 | 35.2 | 18.4 | 27.8 | 39.1 |
| RL Training | StepSearch | - | - | - | 38.6 | 36.6 | **22.6** | 40.0 | - |
| RL Training | **GiGPO** | **46.4** | **64.7** | 46.1 | **41.6** | **43.6** | 18.9 | **68.9** | **47.2** |

Lastly, we observe that the normalization factor $F_{\text{norm}}$ is task-dependent rather than universally helpful. On relatively difficult tasks (such as Look, Pick2, and WebShop), standard-deviation scaling ($F_{\text{norm}} = \text{std}$) could exaggerate gradients from overly difficult samples or highly imbalanced groups, harming update stability; fixing $F_{\text{norm}} = 1$ therefore yields higher success. Yet, $F_{\text{norm}} = 1$ offers no clear advantage on other tasks and both variants perform similarly, which aligns with findings in [72]. This suggests that $F_{\text{norm}} = \text{std}$ can still be beneficial when reward variance is stable.

## 5.3 Performance on QA tasks

As shown in Table 2, GiGPO achieves strong and consistent gains on multi-turn search-augmented QA tasks, reaching 42.1% at 3B and 47.2% at 7B, and significantly outperforming prior strong baselines such as Search-R1 and StepSearch. Although search-augmented QA is relatively short-horizon, the step-level signals of GiGPO still yield meaningful improvements. Furthermore, we observe that GiGPO is markedly more tool-efficient. Under a limit of at most 3 tool calls per query, the 7B model requires only ∼0.9 calls on average for single-hop tasks and ∼1.6 calls on average for multi-hop tasks: matching the superior performance of OTC [61], which achieves ∼1.0 and ∼1.7 calls, respectively, on the same tasks. This efficiency likely stems from GiGPO's ability to effectively identify and suppress redundant queries in multi-turn decision-making. For instance, in repetitive patterns such as query1

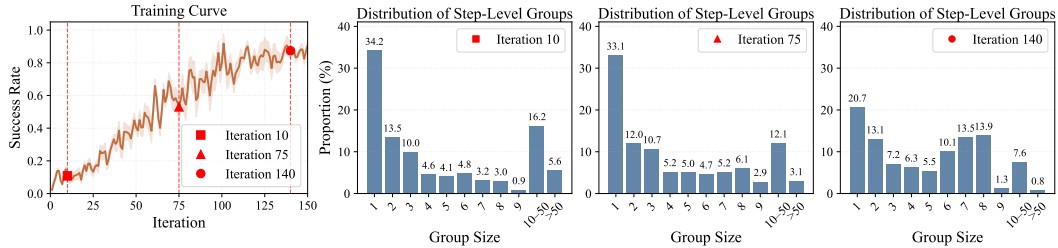

Figure 5: Dynamics of step-level groups during the training in ALFWorld. **Left**: Success rate over training iterations. Vertical red markers denote selected checkpoints (Iterations 10, 75, 140). **Right**: Distribution of step-level group sizes at those checkpoints. For each anchor state $\tilde{s} \in \mathcal{U}$, the group size is given by $|G^S(\tilde{s})|$.

$\rightarrow$ info1 $\rightarrow$ query1 $\rightarrow$ info1 $\rightarrow$ query1 $\rightarrow$ info1, GiGPO groups them into the same step-level group and eliminates unnecessary repeated calls during the training, similar to the example we introduced in Figure 3.

## 5.4 Ablation Study

Next, we conducted an ablation study, comparing $\text{GiGPO}_{\text{w/o std}}$ ($F_{\text{norm}} = 1$), $\text{GiGPO}_{\text{w/ std}}$ ($F_{\text{norm}} = \text{std}$), $\text{GiGPO}_{\text{w/o } A^S}$ (without step relative advantages), and $\text{GiGPO}_{\text{w/o } A^E}$ (without episode relative advantages) to evaluate the impact of each component on performance. We use Qwen2.5-1.5B-Instruct as the policy of the agent. The results are presented in Figure 4.

As illustrated, eliminating either component of the two-level advantage significantly degrades performance. Removing the episode relative advantages ($\text{GiGPO}_{\text{w/o } A^E}$) leads to a substantial drop across all tasks, as the policy no longer receives a stable, trajectory-wide signal to encourage long-range coherence. Similarly, discarding the step relative advantages ($\text{GiGPO}_{\text{w/o } A^S}$) results in pronounced declines, particularly on more complex and demanding tasks such as Cool, Pick2, and Web-Shop, which may require nuanced training feedback at each decision step. In such cases, precise per-step credit assignment is essential for effective learning and policy refinement. Moreover, we can see that the relative performance gap between $\text{GiGPO}_{\text{w/ std}}$ and $\text{GiGPO}_{\text{w/o std}}$

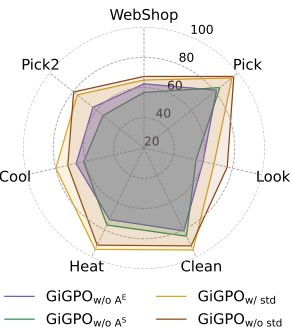

Figure 4: Ablation results. The y-axis shows success rate (%).

is comparatively minor compared to that observed in structural ablations. This suggests that the combination of episode- and step-level signals is the primary driver of performance gains, and that each component is crucial for training LLM agents effectively.

## 5.5 Dynamics of Step-Level Group

In this part, we examine how the distribution of step-level groups evolves throughout training to better understand the utility of GiGPO. We use Qwen2.5-1.5B-Instruct as the base model. We train the LLM agent in ALFWorld and track changes in step-level group sizes throughout training.

As shown in Figure 5, we observe that step-level groups of size 1 (i.e., those with $|G^S(\tilde{s})| = 1$) only account for $< 35\%$ throughout training. This indicates that the majority of states (over 65%) recur across trajectories and therefore contribute to anchor state grouping. Moreover, at iteration 10, large group sizes $|G^S(\tilde{s})| \geq 10$ account for over 20%, reflecting behavioral redundancy in the early stages of training. This is consistent with the fact that immature policies often produce invalid actions or fall into repetitive loops. As training progresses, the group size distribution changes markedly. By iteration 75, we observe a substantial reduction in extreme group sizes: $10 \leq |G^S(\tilde{s})| < 50$ drops from 16.2% to 12.1% and $|G^S(\tilde{s})| \geq 50$ drops from 5.6% to 3.1%. This shift suggests that the agent is learning to avoid previously common dead ends and invalid actions, and is beginning to exhibit more diverse and purposeful decision-making. At iteration 140, the distribution becomes tightly concentrated around group sizes of 6 to 8. Given that $N$ is set to 8, this convergence implies that

all 8 trajectories within an episode group are now behaving consistently: LLM agent has learned a coherent and robust policy for completing the task, aligning with the plateau in success rate ($> 80\%$).

## 5.6 Computational Budget

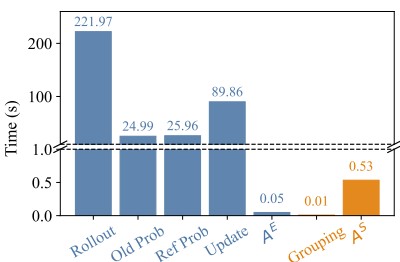

Figure 6: Per-iteration training time breakdown of GiGPO. Blue bars indicate shared components with GRPO. Orange bars show GiGPO-specific additions. The y-axis uses a broken scale to accommodate small values.

Lastly, we analyze the computational budget of GiGPO. GiGPO shares the same core architecture as GRPO, including multi-turn rollouts, computation of old and reference probabilities, and clipped policy updates. Both approaches are critic-free and operate with a single actor LLM, thereby resulting in identical GPU memory usage and LLM rollout costs. The primary additions introduced by GiGPO are the step-relative advantage estimation components, as described in Section 4.2. To evaluate their costs, we train an LLM agent in ALFWorld using Qwen2.5-1.5B-Instruct and record a detailed breakdown of per-iteration training time.

As shown in Figure 6, the additional components incur little to no additional time cost compared to dominant operations such as rollouts, computation of old and reference probabilities, and policy updates, whose total time cost reaches 362.83s per iteration. In contrast, anchor state grouping (involving hashmap lookups) takes only 0.01s per iteration, and the step-relative advantage computation (involving simple arithmetic) adds just 0.53s. These operations account for $< 0.002\%$ of the total per-iteration training time, demonstrating that GiGPO shares the same high computational efficiency as GRPO.

## 6 Conclusions and Limitations

In this work, we proposed GiGPO, a novel group-based RL algorithm to tackle the credit assignment challenge in long-horizon LLM agent training. GiGPO introduces a hierarchical advantage estimation that enables fine-grained per-step credit assignment while retaining the efficiency and stability of group-based RL. By retroactively grouping steps that share the same state across trajectories, it achieves this without incurring additional LLM rollout or GPU memory overhead. Empirical evaluations across complex agentic environments (ALFWorld and WebShop) and search-augmented QA tasks demonstrate that GiGPO significantly outperforms both prompt-based agents and prior RL methods. A potential limitation of GiGPO is its reliance on state matching for anchor group construction. In highly complex environments, identical states may be hard to detect due to noise or subtle differences. Despite this, GiGPO still retains a strong performance lower bound: in the extreme case where no states are repeated across trajectories (i.e., $A^S = 0$), it naturally degrades to GRPO, preserving GRPO's effectiveness and stability in credit assignment. Although this issue is partly mitigated by incorporating similarity-based grouping, exploring more robust state-matching strategies, such as embedding-based representations or domain-specific structural equivalence, remains an important direction.

## Acknowledgements

This research is supported by the Ministry of Education, Singapore, under its Academic Research Fund Tier 1 (RG18/24).

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

# A Open Source Codebase: verl-agent

As part of the new assets released with this work, we propose `verl-agent` (https://github.com/langfengQ/verl-agent), a highly scalable RL training framework for long-horizon, multi-turn LLM agent training.

Our `verl-agent` is built upon the veRL framework [73] and extends it with several features to enable scalable reinforcement learning for long-horizon LLM agents. Key capabilities of our framework include: (**1**) step-wise multi-turn interaction paradigm that avoids concatenating full interaction histories (as in Search-R1 [58] and RAGEN [51]), ensuring efficient memory control and scalability for very long-horizon optimization; (**2**) customizable memory module that allows developers to flexibly determine which historical information to include at each step (e.g., key events, summaries, or external knowledge); (**3**) parallel and group-based environments with a gym-style interface supporting high-throughput rollouts; (**4**) broad model compatibility, including Qwen3 [74], Qwen2.5, and LLaMA3.2, along with LoRA-based fine-tuning [75] for efficient large-model adaptation; (**5**) support for multimodal (vision-language) agents such as Qwen2.5-VL; (**6**) a diverse suite of environments, including Search (tool use), ALFWorld, WebShop, Sokoban, and Gym Cards; (**7**) comprehensive RL algorithm support, encompassing GiGPO, GRPO, PPO, DAPO, RLOO, etc.

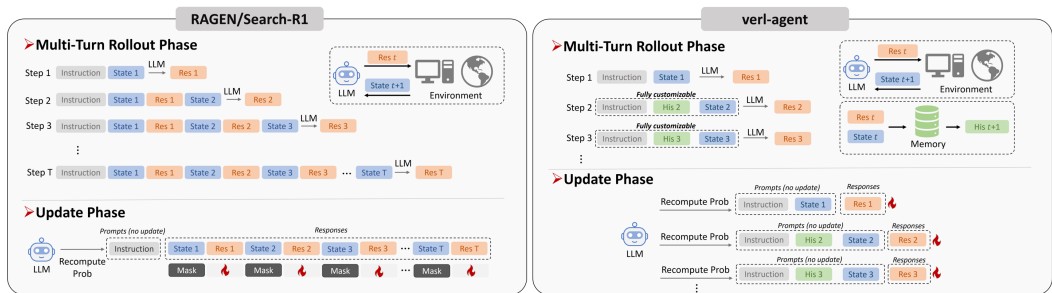

Figure 7: Open-source agentic training framework comparison. **Left**: RAGEN/Search-R1 concatenates the full history at every step, leading to rapidly expanding context. **Right**: `verl-agent` adopts a step-wise multi-turn rollout with flexible per-step input construction and memory control.

# B Broader Impacts

This work introduces an efficient, group-based RL algorithm, GiGPO, that enables fine-grained credit assignment for long-horizon LLM agents. By combining episode-level and step-level advantages, GiGPO improves training stability and agent performance without increasing computational or memory overhead.

The algorithm holds promise for a wide range of applications, including virtual assistants, web automation, educational tools, and embodied AI systems. Its critic-free and scalable design lowers the barrier to training effective multi-step agents, making advanced agent capabilities more accessible to both academic and industrial practitioners. While GiGPO is a methodological contribution with no direct downstream deployment, the improved agent training techniques may indirectly enable more autonomous behavior. As with any advancement in agent capabilities, it is important to consider appropriate safeguards and responsible usage. Overall, GiGPO contributes a valuable tool to the RL and LLM communities, with the potential to accelerate progress in building more intelligent, efficient, and adaptable AI agents.

# C Unbiasedness

We show that setting $F_{\text{norm}} = 1$ leads to an unbiased estimator up to a constant scaling factor. Our derivation follows the approach in [19]. For comparison, the standard REINFORCE Leave-One-Out (RLOO) [16, 17] is defined as:

$$A^{\text{RLOO}}(\tau_i) = R(\tau_i) - \frac{1}{N-1} \sum_{j \neq i} R(\tau_j). \tag{10}$$

We can relate $A^E(\tau_i)$ ($F_{\text{norm}} = 1$) and $A^{\text{RLOO}}(\tau_i)$ by introducing a scaling factor of $\frac{N}{N-1}$:

$$\frac{N}{N-1}A^E(\tau_i) = \frac{N}{N-1}R(\tau_i) - \frac{N}{N-1} \cdot \frac{1}{N}\sum_{j=1}^{N} R(\tau_j) \tag{11}$$

$$= \frac{N}{N-1}R(\tau_i) - \frac{1}{N-1}R(\tau_i) - \frac{1}{N-1}\sum_{j \neq i} R(\tau_j) \tag{12}$$

$$= R(\tau_i) - \frac{1}{N-1}\sum_{j \neq i} R(\tau_j) \tag{13}$$

$$= A^{\text{RLOO}}(\tau_i) \tag{14}$$

Thus, setting $F_{\text{norm}} = 1$ corresponds to a rescaled version of $A^{\text{RLOO}}(\tau_i)$ and scaling the advantage by a constant does not affect the dynamics of policy gradient (it can be absorbed into the learning rate).

## D Pseudo Code

Algorithm 1 summarizes the full GiGPO training procedure. Compared to vanilla GRPO, we highlight the additional parts introduced by GiGPO in italics. In particular, building step-level groups $G^S(\tilde{s})$ is implemented by treating anchor states as keys and aggregating corresponding data into a hash table, which incurs minimal overhead. Furthermore, computing step relative advantages and combining advantages involve only simple arithmetic operations, both of which are lightweight. As such, GiGPO preserves the critic-free, low-memory, and stable convergence properties of group-based RL, while introducing fine-grained credit assignment that is particularly beneficial for training long-horizon LLM agents.

---

**Algorithm 1** Training LLM Agents with GiGPO

---

1: **Require:** Initial policy $\pi_{\theta_{\text{old}}}$, task distribution $p(X)$, discount factor $\gamma$, weighting $\omega$, clipping parameter $\epsilon$, KL penalty $\beta$, group size $N$
2: **for** each training iteration **do**
3:     Update the old policy model: $\theta_{\text{old}} \leftarrow \theta$
4:     // Multi-step rollout phase
5:     Sample task $x \sim p(X)$ and initialize $N$ identical environments
6:     **for** $t = 1$ to $T$ **do**
7:         Sample actions $\{\boldsymbol{a}_t^{(i)} \sim \pi_{\theta_{\text{old}}}(\cdot \mid \boldsymbol{s}_t^{(i)}, x)\}_{i=1}^{N}$
8:         Execute actions, observe rewards $\{r_t^{(i)}\}_{i=1}^{N}$ and next state $\{\boldsymbol{s}_{t+1}^{(i)}\}_{i=1}^{N}$
9:     **end for**
10:     // Grouping phase
11:     Compute episode relative advantages $A^E(\tau_i)$ via Equation (3)
12:     *Build step-level groups $G^S(\tilde{\boldsymbol{s}})$ via the anchor states*
13:     *Compute step relative advantages $A^S(\boldsymbol{a}_t^{(i)})$ via Equation (7)*
14:     // Policy update phase
15:     *Combine advantages: $A(\boldsymbol{a}_t^{(i)}) = A^E(\tau_i) + \omega A^S(\boldsymbol{a}_t^{(i)})$*
16:     Update policy $\theta$ by maximizing objective $\mathcal{J}_{\text{GiGPO}}(\theta)$
17: **end for**

---

## E Experiment Details

### E.1 Details of Training

**Hyperparameters for ALFWorld.** All methods are configured with identical hyperparameters: the maximum prompt length is 2048 tokens, and the maximum response length is 512 tokens. Each episode allows up to 50 environment steps. The learning rate is set to 1e-6 for the actor and 1e-5 for the critic (used only in PPO). We adopt a rule-based reward, assigning a reward of 10 for success and 0 for failure. To handle invalid actions generated by the agent, we apply a reward penalty of

-0.1. For all group-based RL methods, we use a group size of 8 and sample 16 different groups per rollout, resulting in a total of $16 \times 8 = 128$ environments. In contrast, PPO uses 128 separate environments for rollouts. The rollout temperature is set to 1.0, while the validation temperature is set to 0.4. The mini-batch size is 256, and the KL-divergence loss coefficient is set to 0.01. For GiGPO, the weighting coefficient $\omega$ is fixed at 1 without further tuning, and the discount factor $\gamma$ is set to 0.95.

**Hyperparameters for WebShop.** All methods are configured with identical hyperparameters: the maximum prompt length is 4096 tokens, and the maximum response length is 512 tokens. Each episode is limited to 15 environment steps. The learning rate is 1e-6 for the actor and 1e-5 for the critic (used only in PPO). We adopt a rule-based reward, assigning a reward of 10 for success and 0 for failure. Invalid actions are penalized with a reward of -0.1. As with ALFWorld, all group-based RL methods use a group size of 8 and sample 16 groups per rollout, totaling $16 \times 8 = 128$ environments. PPO, on the other hand, uses 128 distinct environments for rollouts. The rollout temperature is set to 1.0, while the validation temperature is set to 0.4. The mini-batch size is 64, and the KL-divergence loss coefficient is set to 0.01. For GiGPO, the weighting coefficient $\omega$ is set to 1 without additional tuning, and the discount factor $\gamma$ is set to 0.95.

**Hyperparameters for Search-Augmented QA.** The maximum prompt length is 4096 tokens, and the maximum response length is 512 tokens. The max turn is set to 4. The learning rate is 1e-6 for the actor. We adopt a rule-based reward, assigning a reward of 1 for success and 0 for failure. Invalid actions are penalized with a reward of -0.01. We set the train data size to 256 and use a group size of 5. Rollout and validation temperatures are set to 1.0 and 0.0, respectively. The mini-batch size is 512, and the KL-divergence loss coefficient is set to 0.001. The weighting coefficient $\omega$ is set to 1 without additional tuning, and the discount factor $\gamma$ is set to 0.95.

**Computing Details.** For ALFWorld and WebShop, Qwen2.5-1.5B experiments are run on 2×H100 GPUs and Qwen2.5-7B on 4×H100 GPUs, each for 150 iterations. For search-augmented QA, Qwen2.5-3B uses 4×H100 GPUs and Qwen2.5-7B uses 8×H100 GPUs, each for 200 iterations.

## E.2 Prompts

The prompts we use for LLM agents are presented in Figure 8, Figure 9, and Figure 10. These prompt templates are constructed using Python-style string formatting, where placeholders enclosed in curly braces ({}) represent semantic slots. These placeholders, such as {task_description}, {step_count}, and {current_observation}, are dynamically populated at runtime via Python's .format() function. To enrich the agent's context, we use historical information and set the history length to 2 for ALFWorld and WebShop and the full history for search-augmented QA experiments.

The <think> </think> block instructs the agent to explicitly perform step-by-step reasoning, thereby promoting chain-of-thought [76] style deliberation. The <action> </action> block is used to clearly indicate the final action decision. The search agent outputs reasoning traces within <think> </think>, issues search queries within <search> </search>, provides anwsers within <anwser> </anwser>. Retrieved evidence from the retriever is presented in <information> </information> tags.

---

**Prompt Template for ALFWorld**

You are an expert agent operating in the ALFRED embodied Environment. Your task is to: {task_description}. Prior to this step, you have already taken {step_count} step(s). Below are the most recent {history_length} observations and the corresponding actions you took: {action_history}. You are now at step {current_step} and your current observation is: {current_observation}. Your admissible actions of the current situation are: [{admissible_actions}].

Now it's your turn to take an action. You should first reason step-by-step about the current situation. This reasoning process MUST be enclosed within <think> </think> tags. Once you've finished your reasoning, you should choose an admissible action for current step and present it within <action> </action> tags.

Figure 8: The prompt template of ALFWorld agents.

> **Prompt Template for WebShop**
>
> You are an expert autonomous agent operating in the WebShop e-commerce environment. Your task is to: {task_description}. Prior to this step, you have already taken {step_count} step(s). Below are the most recent {history_length} observations and the corresponding actions you took: {action_history}. You are now at step {current_step} and your current observation is: {current_observation}. Your admissible actions for the current situation are: [{available_actions}].
> Now it's your turn to take one action for the current step. You should first reason step-by-step about the current situation, then think carefully which admissible action best advances the shopping goal. This reasoning process MUST be enclosed within <think> </think> tags. Once you've finished your reasoning, you should choose an admissible action for current step and present it within <action> </action> tags.

Figure 9: The prompt template used for WebShop agents.

> **Prompt Template for Search**
>
> You are an expert agent tasked with answering the given question step-by-step. Your question: {task_description}. Prior to this step, you have already taken {step_count} step(s). Below is the interaction history where <search> </search> wrapped your past search queries and <information> </information> wrapped the corresponding search results returned by the external search engine. History: {memory_context}
> Now it's your turn to respond for the current step. You should first conduct reasoning process. This process MUST be enclosed within <think> </think> tags. After completing your reasoning, choose only one of the following actions (do not perform both):
> (1) If you find you lack some knowledge, you can call a search engine to get more external information using format: <search> your query </search>.
> (2) If you have enough knowledge to answer the question confidently, provide your final answer within <answer> </answer> tags, without detailed illustrations. For example, <answer>Beijing</answer>.

Figure 10: The prompt template of Search agents.

### E.3 Performance on Vision-Language Agents

We conduct additional experiments in vision-language model (VLM) settings where agents must reason over both visual and textual inputs. We use two interactive game environments: Sokoban [77] with a 6×6 size and EZPoints in Gym Cards [39] as shown in Figure 11. Sokoban is a classic grid-based puzzle game where the agent must push boxes onto designated goal positions. Solving Sokoban requires spatial reasoning and long-term planning. In EZPoints, the agent is presented with two playing cards and must construct an arithmetic formula step-by-step such that the final result equals a target value of 12. This setting emphasizes symbolic reasoning and multi-step planning. In these tasks, the agent

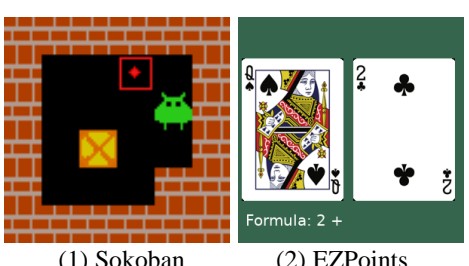

(1) Sokoban      (2) EZPoints

Figure 11: Visual observation of VLM agents in Sokoban and EZPoints.

receives an RGB image representing the current environment state along with a textual instruction, and must take sequential actions to complete the task. All methods are built upon Qwen2.5-VL-3B-Instruct [78].

As shown in Table 3, GiGPO significantly outperforms the prompting baseline and GRPO on both tasks. Notably, GiGPO achieves 81.0% success on Sokoban and 100% success on EZPoints, suggesting its ability to generalize beyond language-only settings.

### E.4 Orthogonality to Single-Turn Group-Based RL

GiGPO remains orthogonal to other advancements in single-turn group-based RL, allowing it to incorporate complementary techniques without loss of generality. To verify this, we consider

Table 3: Success rates (%) of VLM agents using Qwen2.5-VL-3B-Instruct.

| Type | Method | Sokoban [6×6] | EZPoints |
|------|--------|---------------|----------|
| Prompting | Qwen2.5-VL | 11.7 | 3.1 |
| RL Training | GRPO | $67.1_{\pm 4.7}$ | $86.9_{\pm 3.4}$ |
| RL Training | **GiGPO$_{\text{w/ std}}$** | $76.9_{\pm 2.7}$ | **$100.0_{\pm 0.0}$** |
| RL Training | **GiGPO$_{\text{w/o std}}$** | **$81.0_{\pm 3.6}$** | **$100.0_{\pm 0.0}$** |

DAPO [20], which builds on GRPO with dynamic sampling and clip-higher techniques. We integrate both into GiGPO, yielding a variant denoted as GiGPO$_{\text{dynamic}}$.

As shown in Table 4, DAPO improves over GRPO, confirming the benefits of its techniques. Importantly, GiGPO$_{\text{dynamic}}$ further outperforms DAPO, demonstrating that it can effectively benefit from and amplify such improvements, achieving better results (75.0%) on WebShop.

Table 4: Performance on WebShop using Qwen2.5-1.5B-Instruct.

| Type | Method | Score | Success Rate (%) |
|------|--------|-------|------------------|
| Prompting | Qwen2.5 | 23.1 | 5.2 |
| RL Training | GRPO | $75.8_{\pm 3.5}$ | $56.8_{\pm 3.8}$ |
| RL Training | DAPO | $84.6_{\pm 2.9}$ | $66.1_{\pm 3.2}$ |
| RL Training | **GiGPO$_{\text{dynamic}}$** | **$87.5_{\pm 1.6}$** | **$75.0_{\pm 3.5}$** |

### E.5 Sensitivity Analysis on $\omega$

We further analyze GiGPO's sensitivity to the key hyperparameter $\omega$, which balances the episode-level advantage $A^E$ and the step-level advantage $A^S$. We conducted experiments on the WebShop task, using Qwen2.5-1.5B-Instruct.

Table 5: Sensitivity analysis on $\omega$ for the WebShop task.

| $\omega$ | 0.0 | 0.2 | 0.4 | 0.6 | 0.8 | 1.0 | 1.2 | 1.4 |
|----------|-----|-----|-----|-----|-----|-----|-----|-----|
| Score | 76.2 | 79.6 | 82.4 | 83.5 | 84.9 | 83.5 | 82.6 | 77.0 |
| Success Rate (%) | 56.6 | 63.1 | 65.8 | 67.2 | 68.3 | 67.4 | 66.5 | 56.3 |

As shown in Table 5, GiGPO needs an appropriate $\omega$ to work best. Increasing $\omega$ initially improves performance due to the added fine-grained step-level reward. However, performance declines beyond the optimum ($\omega = 0.8$), suggesting that excessive emphasis on step-level signals can suppress useful trajectory-level guidance. Furthermore, GiGPO is relatively insensitive to $\omega$ within the range $[0.4, 1.2]$, demonstrating a reasonable degree of robustness to $\omega$.

# F  Reasoning Behavior

We next present the complete trajectory of multi-turn interaction between agent and environment, and demonstrate the reasoning behavior of LLM agents (Qwen2.5-7B-Instruct) trained by GiGPO.

## F.1  ALFWorld

**Environment (Step 1)**

-= Welcome to TextWorld, ALFRED! =-
Your task is to: heat some egg and put it in countertop. You are in the middle of a room. Looking quickly around you, you see a cabinet 9, a cabinet 8, a cabinet 7, a cabinet 6, a cabinet 5, a cabinet 4, a cabinet 3, a cabinet 2, a cabinet 1, a coffeemachine 1, a countertop 2, a countertop 1, a drawer 13, a drawer 12, a drawer 11, a drawer 10, a drawer 9, a drawer 8, a drawer 7, a drawer 6, a drawer 5, a drawer 4, a drawer 3, a drawer 2, a drawer 1, a fridge 1, a garbagecan 1, a microwave 1, a sinkbasin 1, a stoveburner 6, a stoveburner 5, a stoveburner 4, a stoveburner 3, a stoveburner 2, a stoveburner 1, and a toaster 1.

**Agent (Step 1)**

<think> To heat an egg and put it on the countertop, I need to first find an egg. Given that eggs are commonly stored in the fridge, I should check the fridge first to see if there is an egg available. </think>
<action>go to fridge 1</action>

**Environment (Step 2)**

You arrive at fridge 1. The fridge 1 is closed.

**Agent (Step 2)**

<think>
1. The task is to heat an egg and put it on the countertop.
2. I am currently at the fridge 1, and the fridge is closed. This means I need to open the fridge to check if there is an egg inside.
3. If there is an egg in the fridge, I will take it out, heat it, and then put it on the countertop.
4. If there is no egg in the fridge, I will need to check other locations such as the cabinets, countertop, and garbagecan, but the fridge is the most likely place to find an egg.
5. After opening the fridge and finding the egg, I will need to take it out, heat it using the microwave or stovetop, and then put it on the countertop.
</think>
<action>open fridge 1</action>

**Environment (Step 3)**

You open the fridge 1. The fridge 1 is open. In it, you see a bowl 1, a pan 1, a plate 2, and a potato 1.

**Agent (Step 3)**

<think>
1. The task is to heat an egg and put it on the countertop.
2. I have already opened the fridge 1 and found a bowl 1, a pan 1, a plate 2, and a potato 1 inside. There is no egg in the fridge.
3. Since the egg is not in the fridge, I need to check other locations where an egg might be found, such as the cabinets, countertop, and garbagecan.
4. The countertop is mentioned in the task, so I should check the countertop to see if there is an egg there.
5. If the egg is not found on the countertop, I will check the cabinets and the garbagecan.
</think>
<action>go to countertop 1</action>

**Environment (Step 4)**

You arrive at countertop 1. On the countertop 1, you see a apple 1, a butterknife 2, a dishsponge 1, a fork 1, a lettuce 2, a pan 2, a pot 1, a soapbottle 3, a soapbottle 1, a spoon 2, a tomato 2, a tomato 1, a window 3, and a window 1.

**Agent (Step 4)**

<think>
1. The task is to heat an egg and put it on the countertop.
2. To heat an egg, I need to first find an egg.
3. The most likely places to find an egg would be the fridge or the countertop, as eggs are often stored in cool places to keep them fresh.
4. I have already opened fridge 1 and found it contains a bowl, a pan, a plate, and a potato, but no eggs.
5. The current observation shows that I am at countertop 1, which contains various food items but no eggs.
6. Since I have not yet found the egg, I should check the fridge again in case I missed it, or I should check the countertop 2, as it is another common place to store eggs.
7. After checking the fridge and countertop 1, I should consider the sinkbasin 1, as eggs might be washed there before being used.
8. If I still cannot find the egg, I should check the cabinets, as eggs are sometimes stored in cabinets to keep them organized and out of the way.
</think>
<action>go to countertop 2</action>

**Environment (Step 5)**

You arrive at countertop 2. On the countertop 2, you see a bread 2, a bread 1, a cellphone 1, a egg 3, a egg 2, a egg 1, a fork 2, a kettle 1, a knife 1, a ladle 1, a lettuce 1, a pot 2, a potato 2, a spatula 2, and a tomato 3.

**Agent (Step 5)**

<think>
1. The task is to heat an egg and put it on the countertop.
2. I have already opened fridge 1 and observed its contents, which include a bowl, a pan, a plate, and a potato, but no eggs.
3. The current observation shows that there are eggs on countertop 2. I need to take an egg from countertop 2 to proceed with the task.
4. After taking the egg, I will need to heat it. To heat the egg, I will need to go to the microwave, as it is the most common appliance for heating food.
5. The available actions include taking different eggs from countertop 2, as well as going to the microwave to heat the egg.
</think>
<action>take egg 1 from countertop 2</action>

**Environment (Step 6)**

You pick up the egg 1 from the countertop 2.

**Agent (Step 6)**

<think>
1. I have just picked up the egg 1 from the countertop 2.
2. My next step is to heat the egg. To do this, I need to place the egg in a suitable heating device, such as a microwave or a stovetop.
3. The most convenient and common device for heating an egg is the microwave. Therefore, I will go to the microwave 1 to heat the egg.
4. After heating the egg, I need to put it on the countertop. Since I have already picked up the egg, I will go to the countertop to place the heated egg there.
</think>
<action>go to microwave 1</action>

**Environment (Step 7)**

You arrive at microwave 1. The microwave 1 is closed.

**Agent (Step 7)**

<think>
1. I have just taken the egg 1 from the countertop 2.
2. My current location is microwave 1, and the microwave is closed.
3. To heat the egg, I need to open the microwave to place the egg inside and then heat it.
4. After heating the egg, I need to put it on the countertop.
</think>
<action>open microwave 1</action>

**Environment (Step 8)**

You open the microwave 1. The microwave 1 is open. In it, you see nothing.

**Agent (Step 8)**

<think>
1. I have just opened the microwave 1 and found nothing inside.
2. The egg 1 is on the countertop 2, and I have already gone to the microwave 1 to open it.
3. The next step is to heat the egg 1. Since the egg is on the countertop and I have opened the microwave, I can now heat the egg using the microwave 1.
4. After heating the egg, I need to put it on the countertop. The action 'heat egg 1 with microwave 1' is available, which is the appropriate action to take next.
</think>
<action>heat egg 1 with microwave 1</action>

**Environment (Step 9)**

You heat the egg 1 using the microwave 1.

**Agent (Step 9)**

<think>
1. I have just heated the egg 1 using the microwave 1.
2. My next step is to put the heated egg on the countertop.
3. To do this, I need to go to the countertop where I can place the egg.
4. The admissible actions include 'go to countertop 1' and 'go to countertop 2', which are the appropriate actions to take to reach the countertop.
5. Since I have not specified which countertop to use, I can choose either one. For simplicity, I will choose 'go to countertop 1'.
</think>
<action>go to countertop 1</action>

**Environment (Step 10)**

You arrive at countertop 1. On the countertop 1, you see a apple 1, a butterknife 2, a dishsponge 1, a fork 1, a lettuce 2, a pan 2, a pot 1, a soapbottle 3, a soapbottle 1, a spoon 2, a tomato 2, a tomato 1, a window 3, and a window 1.

**Agent (Step 10)**

<think>
1. I have just heated the egg 1 using the microwave 1.
2. My current location is countertop 1, where I see various kitchen items including a window and a tomato, but no egg.
3. The action 'move egg 1 to countertop 1' is available, which is the appropriate action to take next. This will place the heated egg on the countertop.
</think>
<action>move egg 1 to countertop 1</action>

## F.2  WebShop

**Environment (Step 3)**

Instruction: Find me loose fit, slim fit men's tuxedo shirts with long sleeve, short sleeve, polyester cotton, elastic waist, regular fit for gym workout with color: b-blue, and size: xx-large, and price lower than 40.00 dollars. Your current observation is: 'Search'.

'Back to Search'
'Page 1 (Total results: 50)'
'Next >'
'B09Q67H373': 'JSPOYOU 2022 Newly T-Shirt for Mens Funny 3D Graphics Pattern Crewneck Short Sleeve Tees Big and Tall Summer Casual Comfy Top', '$1.99 to $8.99'
'B09QQP3356': 'HAUKLIE Men's Sports Waffle Ribbed Polo Shirts Summer Short Sleeve Cotton Muscle Quarter-Zip Henley T-Shirt Tunics Tops', '$10.99'
'B09NNMV9LN': 'JSPOYOU Mens Short Sleeve Crewneck 3D Graphic Tunic Shirts Big & Tall Tie Dye Summer Top Basic Designed Classic Cotton Shirt', '$3.99 to $10.99'
'B09S3BN15C': 'Mens Linen Shirt,Men's Striped Shirts Casual Short Sleeve Button Down Shirts Regular Fit Hawaiian Shirts Beach Tees Tops', '$3.78 to $11.38'
'B085WQKRRJ': 'CLANMILUMS Men's Classic Comfort Soft Regular Fit Short Sleeve Henley T-Shirt Tee', '$17.99 to $19.99'
'B078WSND96': 'Clearance Deals Retro T-Shirt Bee Printed Blouse, 2021 Summer Fashion Boys Men 3D Round Neck Short Sleeve Tee Tops', '$9.99 to $11.99'
'B08JD14GJF': 'COOrun Women's UPF 50+ Sun Protection Shirts Quick Dry Long Sleeve Shirts Lightweight T-Shirt Outdoor Hiking Runing Fishing', '$9.98 to $16.98'
'B09T756KQ5': 'WENKOMG1 Men's Crewneck Goth Tee Shirts Skull Print Tops Spring/Summer Long Sleeve Sports T-Shirt Baggy Y2K Soft Streetwear', '$4.09 to $8.09'
'B09R9YCM6R': 'XXBR Summer T-shirts for Mens, Soldier Short Sleeve 3D Street Vintage Printed Shirt Slim Fit Muscle Casual Tee Tops', '$8.98 to $11.99'
'B09Q5ZHRVM': 'Yinimo Mens Gym Shorts Fashionable Loose Beach Drawstring Turkey Print Beach Cool Shorts', '$19.19'.

'Back to Search'
'< Prev'
'color': 'a-yellow' , 'b-blue' , 'c-blue' , 'd-black' , 'd-gray' , 'd-navy'
'size': 'small' , 'medium' , 'large' , 'x-large' , 'xx-large'
'JSPOYOU 2022 Newly T-Shirt for Mens Funny 3D Graphics Pattern Crewneck Short Sleeve Tees Big and Tall Summer Casual Comfy Top'
'Price: $1.99 to $8.99' , 'Rating: N.A.' , 'Description' , 'Features' , 'Reviews' , 'Buy Now'.

**Agent (Step 3)**

<think> Based on the previous observations and actions, it is clear that the search is not yielding any results that match the specific criteria of loose fit, slim fit men's tuxedo shirts with long and short sleeves, regular fit for gym workout, and the desired color (b-blue) and size (xx-large). The current observation shows that the search results are not matching the specific product type (tuxedo shirt) and the specific fit (regular fit for gym workout).

Given the current options, the best action is to refine the search by first checking the color and size options. Since the product type and fit are not matching, I will first click on the color 'b-blue' and then check the size options.

This approach is based on the assumption that the product type and fit might be in the search results but not explicitly labeled as tuxedo shirts or regular fit for gym workout. By clicking on 'b-blue' color, I can narrow down the search and then check the size options to see if the product is a tuxedo shirt and if it is regular fit for gym workout. </think>
<action>click[b-blue]</action>

**Environment (Step 4)**

'Back to Search'
'< Prev'
'color': 'a-yellow' , 'b-blue' , 'c-blue' , 'd-black' , 'd-gray' , 'd-navy'
'size': 'small' , 'medium' , 'large' , 'x-large' , 'xx-large'
'JSPOYOU 2022 Newly T-Shirt for Mens Funny 3D Graphics Pattern Crewneck Short Sleeve Tees Big and Tall Summer Casual Comfy Top'
'Price: $1.99 to $8.99' , 'Rating: N.A.' , 'Description' , 'Features' , 'Reviews' , 'Buy Now'.

**Agent (Step 4)**

<think> The task is to find loose fit, slim fit men's tuxedo shirts with long and short sleeves, polyester cotton, elastic waist, regular fit for gym workout, in blue color (b-blue), size xx-large, and with a price lower than $40.00. From the current observation, it is clear that the search is already filtered to include the color b-blue. The next step is to filter the size to xx-large and check the price range. </think>
<action>click[xx-large]</action>

**Environment (Step 5)**

'Back to Search'
'< Prev'
'color': 'a-yellow' , 'b-blue' , 'c-blue' , 'd-black' , 'd-gray' , 'd-navy'
'size': 'small' , 'medium' , 'large' , 'x-large' , 'xx-large'
'JSPOYOU 2022 Newly T-Shirt for Mens Funny 3D Graphics Pattern Crewneck Short Sleeve Tees Big and Tall Summer Casual Comfy Top'
'Price: $1.99 to $8.99' , 'Rating: N.A.' , 'Description' , 'Features' , 'Reviews' , 'Buy Now'.

**Agent (Step 5)**

<think> The task is to find loose fit, slim fit men's tuxedo shirts with long and short sleeves, made of polyester cotton, with an elastic waist, regular fit for gym workouts, in color b-blue, size xx-large, and with a price lower than $40.00. From the current observation, it is clear that the search is already filtered to include the color b-blue, size xx-large and price lower than $40.00. Therefore, the next step is to click the 'Buy Now' button. </think>
<action>click[buy now]</action>

## F.3  Search

**Environment (Step 1)**

Who is the guy in the video of the performer of One Last Time?

**Agent (Step 1)**

<think> To answer this question, I need to determine who the guy in the video is. The question refers to the performer of the song "One Last Time." The song "One Last Time" was performed by the boy band *NSYNC. The video likely features the members of *NSYNC. Without more context, I cannot determine which specific member is referred to. I will need to search for the video to determine the specific guy. </think>
<search>who is the guy in the video of the performer of One Last Time</search>

**Environment (Step 2)**

<information>{"result": "Doc 1: "Just One Last Time" uploaded to Guetta's official VEVO channel on December 3, 2012. The music video for the track was filmed in Los Angeles during October 2012 by director Colin Tilley. It features a man going in his half-burned house and lying on the bed and remembering the death of his girlfriend, reliving it and trying to change the past in his mind: he is sneaking out of his girlfriend's house to go and play a game of pool with Guetta, only to hear that the house is on fire. He races back to save his girlfriend, this time managing to do so, Doc 2: "One Last Time (Ariana Grande song)" rest of the video then is presented through the use of a handheld video camera. Grande is in a Fiat car as the passenger, with her boyfriend (played by Matt Bennett), who is recording. They are stuck in traffic as they try to leave the city, implied to be Los Angeles, California. While everyone around them is panicking, Grande seems to be calm as she wants to get a closer look at the comet. Impatient, she leaves the car while Bennett tries to stop her, and then follows. They make their way through the crowds, but are stopped by the Doc 3: "One Last Time (Ariana Grande song)" and the United States, where it became Grande's sixth top-twenty single on the US ""Billboard"" Hot 100. The single's peak of number 13 on the US ""Billboard"" Hot 100 ended Grande's streak of consecutive top ten hits from 2014. It was later certified platinum by the Recording Industry Association of America (RIAA) and Australian Recording Industry Association (ARIA). The song's accompanying music video was released on February 15, 2015 and was directed by Max Landis. The music video is visually presented as a found footage, similar to Landis' previous work."}</information>

**Agent (Step 2)**

<think> From the information provided, there are two songs titled "One Last Time" mentioned. The first one is by David Guetta, and the second one is by Ariana Grande. The question asks about the video of the performer of "One Last Time". The information provided for the Ariana Grande song includes details about the music video, which was released on February 15, 2015, directed by Max Landis. The video is described as a found footage. The question might be referring to the video of Ariana Grande's "One Last Time". The information does not provide a specific name for the guy in the video, so I will assume the question is about the video of Ariana Grande's song. </think>
<search>who is the guy in the video of Ariana Grande's One Last Time</search>

<information>{"result": "Doc 1: "One Last Time (Ariana Grande song)" rest of the video then is presented through the use of a handheld video camera. Grande is in a Fiat car as the passenger, with her boyfriend (played by Matt Bennett), who is recording. They are stuck in traffic as they try to leave the city, implied to be Los Angeles, California. While everyone around them is panicking, Grande seems to be calm as she wants to get a closer look at the comet. Impatient, she leaves the car while Bennett tries to stop her, and then follows. They make their way through the crowds, but are stopped by the Doc 2: "One Last Time (Ariana Grande song)"and the United States, where it became Grande's sixth top-twenty single on the US ""Billboard"" Hot 100. The single's peak of number 13 on the US ""Billboard"" Hot 100 ended Grande's streak of consecutive top ten hits from 2014. It was later certified platinum by the Recording Industry Association of America (RIAA) and Australian Recording Industry Association (ARIA). The song's accompanying music video was released on February 15, 2015 and was directed by Max Landis. The music video is visually presented as a found footage, similar to Landis' previous work. The video faced controversy due to its similarity to the Doc 3: "One Last Time (Ariana Grande song)" later dismissed after both sides filed a Joint Stipulation for Dismissal. Grande came up with the idea of the music video and contacted Max Landis, known for his 2012 film ""Chronicle"". In an interview, Grande explained: ""It's very unique, it's very different, I called Max [Landis] up and I said 'hey I want to do something that ends exactly like this' and I described the ending shot to him and then I said I also want it to be (a) one take [music video] 2026 and I want the ending shot to look like this, And I described those two things"}</information>

**Agent (Step 3)**

<think> From the information provided, it seems the question is about the music video for Ariana Grande's song "One Last Time". The video was directed by Max Landis and presented as a found footage. The video shows Ariana Grande and her boyfriend (played by Matt Bennett) in a Fiat car, stuck in traffic trying to leave the city. The "guy" in the video is Matt Bennett, who plays Grande's boyfriend in the video. </think>
<answer>Matt Bennett</answer>

