# OpenReview forum: "Group-in-Group Policy Optimization for LLM Agent Training"
_NeurIPS.cc/2025/Conference — NeurIPS 2025 poster_

### Official Review · Reviewer_fP6w · 2025-06-30

**Clarity:** 3
**Significance:** 3
**Originality:** 3
**Rating:** 5
**Confidence:** 4

**Summary:**

This paper aims to address the credit assignment problem in multi-step reinforcement learning and discovers the anchor state phenomenon. The authors introduce step-level fine-grained advantage computation, which enables better reuse of existing data and combines with episode-level advantages to obtain more accurate advantage signals. The experimental results are impressive, demonstrating significant performance improvements.

**Questions:**

During the actual rollout process, in each turn, how are $r_t^{(i)}$ and $r_k^{(i)}$ given? My understanding is that only the final step receives a reward, while all other steps have a reward of 0?

**Ethical Concerns:**

["NO or VERY MINOR ethics concerns only"]

**Final Justification:**

The authors have supplemented some experiments, which addressed my main concerns, if integrated into the revised paper, can meet the acceptance standards.

**Limitations:**

1. The method is only applicable to environments with limited or simple state spaces where repetitive environmental states are easily identifiable or frequently occur.
2. The method incurs high training overhead and requires substantial computational resources.

**Quality:**

3

**Strengths And Weaknesses:**

Strengths:
1. High writing quality and clarity: The paper is well-written and easy to understand, with well-formatted figures, tables, and equations that enhance comprehension.
2. Accurate problem identification with strong motivation: The paper effectively targets the credit assignment problem in multi-step reinforcement learning with well-justified motivation.
3. Clever method design: The authors first discover the anchor state phenomenon during experimentation, then introduce step-level advantage computation to fully utilize this data information, incorporating more fine-grained advantage signals. The computational overhead is negligible, requiring no additional rollouts or complex calculations.
4. Impressive experimental results: The results demonstrate the effectiveness of the proposed method design.

Weaknesses:
1. Method design relies on task characteristics, raising concerns about generalizability: The method depends on identical states to construct anchor groups, which may only be applicable to environments where identical states can be detected. For more complex and open multi-step environments (such as multi-turn dialogue, web queries, textgame: SOTOPIA[1], GAIA[2], NegotiationArena[3], LMRL Gym[4]), this method may not be applicable. The two environments validated in the paper, ALFWorld and WebShop, are relatively simple with limited state spaces, making it easy for identical states to reappear during multi-step interactions.
2. Limited hyperparameter sensitivity analysis: While I understand that multiple experiments are computationally expensive, conducting experiments on the weight parameter w to explore hyperparameter sensitivity would make the work more solid.
3. Missing LRM baselines: Since the method employs the thinking-action paradigm, it should also compare against some LRMs such as DeepSeek-R1.
4. Missing classic multi-step RL baselines: It is recommended to include classic baselines for multi-turn RL such as ArCHer[5] and Raven[6]. Raven may be recent work within three months before submission, so it's not mandatory if conditions don't permit.
5. High computational resource requirements, difficult to reproduce: Based on the submitted code and my experience, the proposed approach requires substantial CPU (at least 64 cores) and GPU resources, resulting in significant computational overhead.

[1] Zhou X, Zhu H, Mathur L, et al. SOTOPIA: Interactive Evaluation for Social Intelligence in Language Agents[C]//The Twelfth International Conference on Learning Representations.

[2] Mialon G, Fourrier C, Wolf T, et al. Gaia: a benchmark for general ai assistants[C]//The Twelfth International Conference on Learning Representations. 2023.

[3] Abdulhai M, White I, Snell C, et al. Lmrl gym: Benchmarks for multi-turn reinforcement learning with language models[J]. arXiv preprint arXiv:2311.18232, 2023.

[4] Bianchi F, Chia P J, Yuksekgonul M, et al. How well can LLMs negotiate? NEGOTIATIONARENA platform and analysis[C]//Proceedings of the 41st International Conference on Machine Learning. 2024: 3935-3951.

[5] Zhou Y, Zanette A, Pan J, et al. ArCHer: Training Language Model Agents via Hierarchical Multi-Turn RL[C]//International Conference on Machine Learning. PMLR, 2024: 62178-62209.

[6] Wang Z, Wang K, Wang Q, et al. Ragen: Understanding self-evolution in llm agents via multi-turn reinforcement learning[J]. arXiv preprint arXiv:2504.20073, 2025.

---

> ### Author Rebuttal · Authors · 2025-07-31
>
> We sincerely appreciate your valuable comments and your positive remarks about our paper’s clear writing, strong motivation with accurate problem identification, and the effective design based on the anchor state grouping.
> We're also glad you found the step-level advantage computation efficient and the experimental results impressive.
> Regarding your questions, we provide our responses below:
>
> > (Weakness 1) Generalizability to complex and open multi-step environments.
>
> Thanks for your valuable insight. As discussed in Sec. 6 (Conclusions and Limitations), we can use **state similarity** to better capture structurally equivalent states in complex and open scenarios.
> We have explored this perspective and conducted the following **two** experiments to evaluate how GiGPO performs under realistic noise where exact state matches are rare. We find **using similarity for GiGPO works well**.
>
> 1. **Experiment 1**
> 	- (Noisy WebShop): We used GPT-4o to generate 551 diverse advertisements, which are **randomly inserted** into the states returned by WebShop to **simulate noise from ad pop-ups**. This substantially **increases the state spaces**.
> 	- (Similarity‐based grouping): We use Ratcliff–Obershelp similarity (ranges from 0 to 1) to construct step-level groups: **grouping the states whose similarity** $\geq$ `sim_thresh`. Exact match is the special case `sim_thresh=1.0`.
>
> 	The results are shown as follows:
>
> 	||Score|Success Rate|Avg. Step Group Size|
> 	|:-:|:-:|:-:|:-:|
> 	|GiGPO (sim_thresh=1.0)|74.5|56.7|1.01|
> 	|GiGPO (sim_thresh=0.9)|82.6|66.0|4.50|
> 	|GiGPO (sim_thresh=0.8)|83.2|66.2|4.96|
> 	|GiGPO (sim_thresh=0.5)|80.7|64.8|6.69|
> 	|GiGPO (sim_thresh=0.2)|78.7|58.6|12.15|
>
> 	We can see that:
> 	- **Exact-match failure**. With `sim_thresh=1.0`, the avg. step group size collapses to $\approx 1$, indicating that almost no identical states are found. As a result, GiGPO **degrades to GRPO** and performance drops.
> 	- **GiGPO remains robust and effective when using similarity-based grouping** even under noisy conditions. As shown, `sim_thresh=0.9/0.8` works well in such a setting. As `sim_thresh` decreases further (e.g., 0.5 or 0.2), group size increases, but similarity quality diminishes, revealing a trade-off between group generalization and precision.
>
> 2. **Experiment 2**
> 	- To further demonstrate GiGPO's applicability in open environments, we applied it to the **Search-R1 environment[1] (web query task)**, where the agent interacts with a search engine (tool-calling) to answer questions.
> 	- We tested both GRPO and GiGPO using Qwen2.5-1.5B-Instruct.
> 	- Since search results are **inherently variable, with extremely large state spaces**, we used similarity-based grouping for GiGPO.
>
> 	||NQ|TriviaQA|PopQA|HotpotQA|2wiki|Musique|Bamboogle|Avg.|
> 	|-|:-:|:-:|:-:|:-:|:-:|:-:|:-:|:-:|
> 	|Search-R1 (Qwen2.5-3B-Instruct)|0.341|0.545|0.378|0.324|0.319|0.103|0.264|0.325|
> 	|GRPO (ours)|0.380|0.541|0.423|0.303|0.269|0.088|0.317|0.331|
> 	|GiGPO (ours)|0.406|0.574|0.441|0.294|0.284|0.103|0.312|0.345|
>
> 	We can see that, even in this open setting, GiGPO's step-level signal provides benefits over GRPO, demonstrating its effectiveness in open agentic tasks.
>
> > (Weakness 2) Hyperparameter sensitivity analysis.
>
> Thanks for your valuable suggestion. We conduct the sensitivity study on the hyperparameter $\omega$ in WebShop.
> | $\omega$           | 0.0  | 0.2  | 0.4  | 0.6  | 0.8  | 1.0  | 1.2  | 1.4  |
> |--------------------|------|------|------|------|------|------|------|------|
> | Score              | 76.2 | 79.6 | 82.4 | 83.5 | 84.9 | 83.5 | 82.6 | 77.0 |
> | Success Rate | 56.6 | 63.1 | 65.8 | 67.2 | 68.3 | 67.4 | 66.5 | 56.3 |
>
> We find that
> - **GiGPO needs an appropriate $\omega$ to work best**.
> - Increasing $\omega$ **initially improves** performance due to the added fine-grained step-level reward. However, performance **declines beyond the optimum** ($\omega=0.8$), suggesting that excessive emphasis on step-level signals can suppress useful trajectory-level guidance.
> - GiGPO is **relatively insensitive** to $\omega$ within the range $[0.4, 1.2]$, demonstrating a reasonable degree of robustness to $\omega$.
>
> > (Weakness 3) LRM baselines.
>
> We conduct experiments for `DeepSeek-R1-0528` and the results are provided below:
>
> ||Pick|Look|Clean|Heat|Cool|Pick2|All|WebShop Score|WebShop Succ.|
> |:-:|:-:|:-:|:-:|:-:|:-:|:-:|:-:|:-:|:-:|
> |DeepSeek-R1|64.2|52.1|46.5|47.0|54.3|55.2|54.5|42.2|33.6|
>
> Compared with Table 1 in the paper, DeepSeek-R1 achieves performance between GPT-4o and Gemini-2.5-Pro across most tasks.
>
> > (Weakness 4) Classic multi-step RL baselines.
>
> Thanks for your valuable suggestion. We conduct experiments for RAGEN[2] on WebShop and the results are provided below:
>
> ||Score|Success Rate|
> |-|-|:-:|
> |RAGEN (Qwen2.5-1.5B-Instruct)|79.7|59.6|
> |GiGPO (Qwen2.5-1.5B-Instruct)|83.5|67.4|
>
> GiGPO shows improvements over RAGEN, further supporting its effectiveness in multi-step settings.
>
> > (Weakness 5) High computational resource requirements.
>
> We appreciate the concern, but we would like to clarify that the cited resource demand is **ONLY the code-level optimization choices**, which are **fully independent of the GiGPO's algorithmic core** and can be adjusted based on available hardware.
>
> - In our experiments, we used 2 H100 GPUs and 64 CPUs **only to shorten wall-clock time ($\approx$ 9h) via high parallelism**. This setup is an implementation convenience, not an algorithmic requirement.
> - The same training can be run on 1 H100 GPU + 32 CPUs (or even smaller setups) by, for example, leveraging the accumulation steps or allowing workers to execute sequentially. This change lengthens training time ($\approx$ 18h) but leaves results unchanged.
>
>
> > (Question 1) How are $r_t^{(i)}$ and $r_k^{(i)}$ given?
>
> - This depends on the specific task and environment. Some environments provide dense rewards, while others are sparse-reward scenarios. In the case of **sparse-reward environments**, your understanding is correct: $r_t^{(i)} = 0$ for $t < T$; $r_T^{(i)} = 1$ if the task is successful, and $0$ otherwise.
> - However, it is worth noting that when constructing step-level groups, we do **not** directly use the raw per-step reward $r_t^{(i)}$ from the environment. Instead, we use the **discounted return** $R_t^{(i)}$, as defined in Equations (5) and (6). This allows us to assign **non-zero values to earlier steps** ($t<T$) and its effectiveness is illustrated in Figure 3 of the main paper.
>
>
> Finally, thank you again for your valuable and encouraging feedback. We will incorporate these clarifications and improvements into our revision.
>
> ---
> References:
>
> [1] Jin, Bowen, et al. Search-R1: Training LLMs to Reason and Leverage Search Engines with Reinforcement Learning. arXiv, 2025.
>
> [2] Wang, Zihan, et al. RAGEN: Understanding Self-Evolution in LLM Agents via Multi-Turn Reinforcement Learning. arXiv, 2025.

---

> > ### Comment · Reviewer_fP6w · 2025-08-05
> > **Response to Authors**
> >
> > Thanks for your detailed responses which have addressed my concerns. I would like to raise my score.

---

> > > ### Author Response · Authors · 2025-08-05
> > > **Thanks to Reviewer fP6w**
> > >
> > > Thank you very much for your positive update. We sincerely appreciate the time and effort you devoted to reviewing our paper!

---

### Official Review · Reviewer_aEAm · 2025-07-02

**Clarity:** 3
**Significance:** 2
**Originality:** 3
**Rating:** 4
**Confidence:** 4

**Summary:**

The paper introduces Group-in-Group Policy Optimization (GiGPO), a novel reinforcement learning technique for training large language model (LLM) agents in long-horizon tasks. GiGPO uses a two-level advantage estimation structure: episode-level advantages for overall trajectory quality (original GRPO) and step-level advantages for fine-grained credit assignment (based on anchor state). This hierarchical approach enables precise optimization without additional rollouts or auxiliary models, maintaining efficiency and scalability. Evaluations on ALFWorld and WebShop show performance gains over existing methods, with over 12% improvement on ALFWorld and over 9% on WebShop.

**Questions:**

1. I think Section 5.5 of the paper is too obvious, and the content of E.3 or E.4 might be more important. How do the authors comment on this?
2. Is it possible to further enhance the technical section of the paper, perhaps by including at least one task demonstrating GiGPO's effectiveness under non-precise anchor state matching?
3. Can the improvements proposed by the author be applied to PPO?
4. Could the experiments be supplemented by applying similar advantage calculations in other reinforcement learning frameworks to strengthen the generality of this idea?

**Ethical Concerns:**

["NO or VERY MINOR ethics concerns only"]

**Limitations:**

yes

**Quality:**

3

**Strengths And Weaknesses:**

Strengths
* The method proposed in this paper is simple to experiment with and intuitively useful.
* The paper is well-written with clear exposition and reasonable experiments.
* This paper reports good performance for ALFWorld and WebShop.

Weaknesses
* As mentioned in the paper, state matching itself is a challenging problem, and the current algorithm is mainly tested on tasks with a limited number of states. This leads to two issues: first, when reading the paper's introduction, I mistakenly thought this was a universal GRPO improvement, and I believe the writing here is a bit flawed; second, the technical depth discussed in the paper is relatively average.

---

> ### Author Rebuttal · Authors · 2025-07-31
>
> Thank you very much for your constructive review and positive feedback.
> We truly appreciate you highlighting the strengths of our method: the simplicity and intuitive design that make it easy to experiment with, the clarity and structure of the paper, and the strong empirical performance.
> Below, we provide our detailed responses to your remaining questions.
>
> > (Weakness 1) Universal GRPO improvement.
>
> We appreciate you pointing out this confusion. To clarify: GiGPO is **not intended for single-turn tasks** (e.g., math or question answering), but is designed as a **universal GRPO improvement for LLM agent scenarios**, which involve multi-turn or multi-step interactions.
> We will revise the text to reflect the intended scope and applicability of GiGPO more accurately.
>
> > (Question 1) Section E.3 or E.4 might be more important.
>
> Thank you for this helpful suggestion. We agree that Sections E.3 and E.4, which demonstrate the generality of GiGPO, offer important insights and may be more impactful than Section 5.5. We will consider moving them into the main paper to better highlight the broader applicability of GiGPO.
>
> > (Question 2) Demonstrating non-precise anchor state matching.
>
> Thanks for your valuable suggestion.
> We have explored this perspective and conducted experiments to evaluate how GiGPO performs under realistic noise where exact state matches are rare. We find **using similarity for GiGPO works well**.
>
> Specifically:
> - (Noisy WebShop): To simulate realistic noise, we used GPT-4o to generate 551 diverse advertisements, which are **randomly inserted** into the states returned by WebShop to **simulate noise from ad pop-ups**. This substantially increases the state spaces.
> - (Similarity‐based grouping): We use Ratcliff–Obershelp similarity (ranges from 0 to 1) to construct step-level groups: **grouping the states whose similarity** $\geq$ `sim_thresh`. Exact match is the special case `sim_thresh=1.0`.
>
> The results are shown as follows:
>
> ||Score|Success Rate|Avg. Step Group Size|
> |:-:|:-:|:-:|:-:|
> |GiGPO (sim_thresh=1.0)|74.5|56.7|1.01|
> |GiGPO (sim_thresh=0.9)|82.6|66.0|4.50|
> |GiGPO (sim_thresh=0.8)|83.2|66.2|4.96|
> |GiGPO (sim_thresh=0.5)|80.7|64.8|6.69|
> |GiGPO (sim_thresh=0.2)|78.7|58.6|12.15|
>
> We can see that:
> - **Exact-match failure**. With `sim_thresh=1.0`, the avg. step group size collapses to $\approx 1$, indicating that almost no identical states are found. As a result, GiGPO **degrades to GRPO** and performance drops.
> - **GiGPO remains robust and effective when using similarity-based grouping** even under noisy conditions. As shown, `sim_thresh=0.9/0.8` works well in such a setting. As `sim_thresh` decreases further (e.g., 0.5 or 0.2), group size increases, but similarity quality diminishes, revealing a trade-off between group generalization and precision.
>
> > (Question 3 & Question 4) Can similar advantage calculations be applied to other RL frameworks?
>
> 1. **Yes**. As discussed in Section E.4, the core ideas of GiGPO (anchor state grouping & step-level advantage) are **general** and can be applied to **all group-based RL frameworks**.
> We have already extended GiGPO to DAPO in Section E.4 and additionally applied it to RLOO for this response. Results are summarized below:
> ||Score|Success Rate|
> |-|-|:-:|
> |DAPO (Sec. E.4)|84.6|66.1|
> |DAPO + Step-Level Adv. (Sec. E.4)|87.5|75.0|
> |RLOO|73.9|52.1|
> |RLOO + Step-Level Adv. |83.3|63.2|
>
> 2. Regarding PPO: As PPO is not a group-based RL method and does not require multiple rollouts for the same task, applying anchor state grouping to PPO is **not straightforward**. However, one potential way is to **introduce task grouping into PPO by rolling out a set of trajectories under the same task**. In such a setting, the core idea of GiGPO can be incorporated into PPO.
>
>
> Thank you again for your valuable feedback. We will incorporate the above points in our revised version to enhance the technical contributions.

---

> > ### Comment · Reviewer_aEAm · 2025-08-04
> >
> > The author's level of detail in responding exceeded expectations, and I am inclined to accept the paper.

---

> > > ### Author Response · Authors · 2025-08-05
> > > **Thanks to Reviewer aEAm**
> > >
> > > Thank you very much for the positive feedback and kind words regarding the details of our responses. We are delighted to see that our reply has addressed all your concerns and sincerely hope this might encourage a positive update of your initial assessment within the review panel.
> > >
> > > As the discussion period has been extended to 8 Aug, there is still ample time for further exchange. Should you have any additional questions or reservations about our work, we are more than willing to engage in further discussion with you.

---

> > > > ### Comment · Reviewer_aEAm · 2025-08-05
> > > >
> > > > I have a technical question that is not very clear to me. What specific tasks can GiGPO be applied to? What other tasks have the potential to use the framework proposed by GiGPO?

---

> ### Author Response · Authors · 2025-08-05
>
> Thank you very much for your reply. Regarding your questions, we provide our responses below.
>
> > What specific tasks can GiGPO be applied to?
>
> GiGPO can be applied to **all LLM agentic tasks**, which typically involve:
> 1. **Interaction with external tools/environments**: Not just generating text — it uses tools, checks websites, calls APIs, operates a computer/phone, or works inside a simulated environment.
> 2. **A multi-step process**: The model doesn't finish in one go (“instruction → answer”) — it takes a step, looks at the result from external environments, decides the next move, and keeps going until the goal is met (“instruction → intermediate steps → answer/goal”).
>
> Examples of LLM agentic tasks to which GiGPO can be readily applied:
> | Task | Interaction Type | Step Example |Reference|
> |-|-|-|-|
> | **Mobile app control** | LLM ↔ Mobile App | Open app → Navigate menus → Perform desired actions → Confirm operation |[1][2]|
> | **Coding & debugging** | LLM ↔ IDE / Compiler | Write code → Run it → See errors → Fix → Run again → ... → Finish |[3][4]|
> | **Math problem solving** | LLM ↔ Calculator | Break down problem → Send formula 1 → Get result 1 → Send formula 2 → Get result 2 → Present final answer |[5][6] |
> | **Web search** | LLM ↔ Search Engine | Take user query → Search the web → Read multiple pages → Search again → ... → Summarize and answer |[7][8][9]|
> | **Robotic control** | LLM ↔ Robot Arm | Receive instructions → Plan path → Move arm → Adjust via sensor feedback → ... → Complete task |[10][11]|
> | **Computer use automation** | LLM ↔ Desktop OS | Open programs → Navigate menus → Copy files → Change settings → Close programs |[12] |
>
> > What other tasks have the potential to use the framework proposed by GiGPO?
>
> Beyond agentic tasks, **multi-turn dialogue** is a natural fit. Multi-turn conversations between a user and an LLM are inherently multi-step, and the “user” can be treated as an “external environment”.
> This makes GiGPO a potential candidate for multi-turn dialogue optimization.
>
> ---
> References:
>
> [1] C Zhang et al. AppAgent: Multimodal Agents as Smartphone Users, CHI 2025.
>
> [2] J Wang et al. Mobile-Agent-v2: Mobile Device Operation Assistant with Effective Navigation via Multi-Agent Collaboration, NeurIPS 2024.
>
> [3] K Zhang et al. CodeAgent: Enhancing Code Generation with Tool-Integrated Agent Systems for Real-World Repo-level Coding Challenges, ACL 2024.
>
> [4] J Yang et al. SWE-agent: Agent-Computer Interfaces Enable Automated Software Engineering, NeurIPS 2024.
>
> [5] Z Gou et al. ToRA: A Tool-Integrated Reasoning Agent for Mathematical Problem Solving, ICLR 2024.
>
> [6] Z Xue et al. SimpleTIR: End-to-End Reinforcement Learning for Multi-Turn Tool-Integrated Reasoning, 2025
>
> [7] OpenAI. Introducing ChatGPT Search, 2024.
>
> [8] B Jin et al. Search-R1: Training LLMs to Reason and Leverage Search Engines with Reinforcement Learning, arXiv 2025.
>
> [9] M Chen et al. ReSearch: Learning to Reason with Search for LLMs via Reinforcement Learning, arXiv 2025.
>
> [10] MJ Kim et al. OpenVLA: An Open-Source Vision-Language-Action Model, CoRL 2024.
>
> [11] G Lu et al. VLA-RL: Towards Masterful and General Robotic Manipulation with Scalable Reinforcement Learning, arXiv 2025.
>
> [12] T Xie et al. OSWorld: Benchmarking Multimodal Agents for Open-Ended Tasks in Real Computer Environments, NeurIPS 2024.

---

### Official Review · Reviewer_vKcT · 2025-07-02

**Clarity:** 3
**Significance:** 3
**Originality:** 4
**Rating:** 4
**Confidence:** 4

**Summary:**

The paper extends the group-based reinforcement learning method from single-turn tasks to long-horizon multi-turn agent training and proposes a Group-in-Group Policy Optimization method, named GiGPO. The method introduces two-level structure to achieve fine-grained credit assignment for LLM agents. At episode level, GiGPO samples a group of trajectories and compute advantage following vanilla GRPO. At step level, GiGPO splits actions into groups based on anchor states, enabling step-level guidance for long-horizon agent refinement.  Experiements on ALFWorld and WebShop with two base models demonstrates the effectiveness of GiGPO.

**Questions:**

See weaknesses.

**Ethical Concerns:**

["NO or VERY MINOR ethics concerns only"]

**Final Justification:**

Most of my questions and concerns have been addressed during the rebuttal, I will keep the score and recommend the acceptance of the paper.

**Limitations:**

The authors discuss their potential limitations in their paper.

**Paper Formatting Concerns:**

There's no formatting issues.

**Quality:**

3

**Strengths And Weaknesses:**

**Strengths**
1. The paper extends the group-based RL method to multi-turn tasks and proposes GiGPO, which designs rewards from both solution-level and step-level, enabling both global and local credit assignment. This significantly improves the agent’s performance over long decision sequences.
2. The step-level credit assignment is constructed retroactively using existing trajectories, thereby avoiding additional rollouts or costly value model estimations, which helps preserving the strengths of group-based RL methods.
3. Experimental results demonstrate that GiGPO achieves substantial improvements over existing baselines across both benchmarks and model scales.
4. The paper is clearly written and easy to follow.


**Weaknesses**
1. Although GiGPO is designed to minimize additional computational overhead, the method requires constructing step-level groups to estimate relative advantages. This process may demand a substantial amount of data to ensure group diversity and representativeness.
2. GiGPO is primarily designed for long-horizon tasks and relies on the presence of anchor states. This design limits its applicability in certain scenarios—for example, in short-horizon or simple tasks, its advantages may not be as significant.
3. GiGPO builds step-level groups by identifying and grouping repeated environment states across trajectories. This requires that environment states be repeatable and comparable across different rollouts. However, in real-world settings, environment states may suffer from noise, variability, or observation inaccuracies, which could hinder accurate step-level grouping and affect the reliability of relative advantage estimation.

---

> ### Author Rebuttal · Authors · 2025-07-31
>
> Thank you very much for your detailed and thoughtful review.
> We truly appreciate you highlighting our credit assignment design as effective and pointing out that GiGPO enables fine-grained guidance while avoiding additional rollouts or costly value model estimations.
> We also thank you for your kind comments on the clarity of the paper and the substantial performance gains across benchmarks and model scales.
> Regarding your questions, we provide our responses as follows.
>
> > (Weakness 1) Constructing step-level groups demands a substantial amount of data.
>
> We agree that increasing data volume can lead to more robust step-level groups. However, we would like to clarify that our experiments on GiGPO were actually conducted using **relatively small data settings** `(batch_size=16, group_n=8)`.
> For comparison, prior agentic training work Search-R1[1] uses `(batch_size=512, group_n=5)`.
>
> Despite this, GiGPO was able to form effective step-level groups and maintain stable training, demonstrating its **efficiency and practicality even under small data conditions**.
>
> > (Weakness 2) Short-horizon or simple tasks.
>
> Thanks for your valuable insight. To explore this, we evaluated GiGPO on a short-horizon agentic task using the **Search-R1 environment[1]**:
>
> Specifically:
> - This task involves answering questions via **search engine tool-calling**, with a maximum of 4 steps (short-horizon).
> - We tested both GRPO and GiGPO using Qwen2.5-1.5B-Instruct.
> - Since search results are inherently variable, with extremely large state spaces, we used similarity-based grouping for GiGPO as introduced in our response to (Weakness 3).
>
> The results are shown as follows (*Note: The “Search-R1” results are sourced from the original paper [1]*):
> ||NQ|TriviaQA|PopQA|HotpotQA|2wiki|Musique|Bamboogle|Avg.|
> |-|:-:|:-:|:-:|:-:|:-:|:-:|:-:|:-:|
> |Search-R1 (Qwen2.5-3B-Instruct)|0.341|0.545|0.378|0.324|0.319|0.103|0.264|0.325|
> |GRPO (ours)|0.380|0.541|0.423|0.303|0.269|0.088|0.317|0.331|
> |GiGPO (ours)|0.406|0.574|0.441|0.294|0.284|0.103|0.312|0.345|
>
> We can see that, despite the short-horizon nature of the task, GiGPO's step-level signals still provided **meaningful improvements** over GRPO. Naturally, this advantage becomes more pronounced as the number of steps increases.
>
> > (Weakness 3) State noise and variability.
>
> Thanks for your valuable insight. As discussed in Sec. 6 (Conclusions and Limitations), we can use **state similarity** to better capture structurally equivalent states in noisy scenarios.
> We have explored this perspective and conducted experiments to evaluate how GiGPO performs under realistic noise where exact state matches are rare. We find **using similarity for GiGPO works well**.
>
> Specifically:
> - (Noisy WebShop): To simulate realistic noise, we used GPT-4o to generate 551 diverse advertisements, which are **randomly inserted** into the states returned by WebShop to **simulate noise from ad pop-ups**. This substantially increases the state spaces.
> - (Similarity‐based grouping): We use Ratcliff–Obershelp similarity (ranges from 0 to 1) to construct step-level groups: **grouping the states whose similarity** $\geq$ `sim_thresh`. Exact match is the special case `sim_thresh=1.0`.
>
> The results are shown as follows:
>
> ||Score|Success Rate|Avg. Step Group Size|
> |:-:|:-:|:-:|:-:|
> |GiGPO (sim_thresh=1.0)|74.5|56.7|1.01|
> |GiGPO (sim_thresh=0.9)|82.6|66.0|4.50|
> |GiGPO (sim_thresh=0.8)|83.2|66.2|4.96|
> |GiGPO (sim_thresh=0.5)|80.7|64.8|6.69|
> |GiGPO (sim_thresh=0.2)|78.7|58.6|12.15|
>
> We can see that:
> - **Exact-match failure**. With `sim_thresh=1.0`, the avg. step group size collapses to $\approx 1$, indicating that almost no identical states are found. As a result, GiGPO **degrades to GRPO** and performance drops.
> - **GiGPO remains robust and effective when using similarity-based grouping** even under noisy conditions. As shown, `sim_thresh=0.9/0.8` works well in such a setting. As `sim_thresh` decreases further (e.g., 0.5 or 0.2), group size increases, but similarity quality diminishes, revealing a trade-off between group generalization and precision.
>
> Thank you again for your valuable and encouraging feedback. We will incorporate all these clarifications and improvements into our revision.
>
> ---
> References:
>
> [1] Jin, Bowen, et al. Search-R1: Training LLMs to Reason and Leverage Search Engines with Reinforcement Learning. arXiv, 2025.

---

> > ### Comment · Reviewer_vKcT · 2025-08-04
> >
> > Thanks for the author's detailed responses and I would like to keep my score.

---

> > > ### Author Response · Authors · 2025-08-05
> > > **Thanks to Reviewer vKcT**
> > >
> > > Thank you very much for your positive feedback and in-depth suggestions. We sincerely hope our detailed responses have adequately addressed all your concerns. As the discussion period has been extended to 8 Aug, there is still ample opportunity for further exchange. If you have any remaining questions or reservations about our work, we would be more than happy to continue the discussion with you.

---

### Official Review · Reviewer_yaqn · 2025-07-03

**Clarity:** 3
**Significance:** 3
**Originality:** 3
**Rating:** 4
**Confidence:** 3

**Summary:**

The authors present GiGPO, a new RL algorithm aimed at tackling the tricky problem of credit assignment for LLM agents in long-horizon tasks. They argue that standard group-based methods like GRPO fail to distinguish good from bad actions within a single long trajectory. The key idea here is to supplement the usual episode-level advantage with a more fine-grained, step-level signal. This is achieved through what they call "anchor state grouping"—a  process that groups together actions taken from the exact same state across different rollouts. This allows for a localized relative advantage calculation without needing extra models or environment interactions. The authors demonstrate on ALFWorld and WebShop that their method provides significant gains over existing approaches while remaining just as efficient as the GRPO baseline.

**Questions:**

1. Equation (8) introduces the hyperparameter ω to trade off the global, episode-level signal (AE) against the local, step-level signal (AS). In the manuscript, ω is fixed at 1 without any tuning or ablation. Could you elaborate on how sensitive the reported results are to this choice? In particular, might certain values of ω shift the optimal solution of the original problem or induce undesirable agent behaviours?  If there are some theoretical analysis, it will be helpful.
2. The proposed anchor-state grouping relies on exact state matches, a condition that may be fragile in noisy or high-dimensional environments. Have you analysed how performance varies with increasing state-space noise or complexity? For example, how does the performance gap between GiGPO and GRPO evolve as the probability of encountering an exact state match diminishes? Any empirical or theoretical insights into this degradation would strengthen the contribution.

**Ethical Concerns:**

["NO or VERY MINOR ethics concerns only"]

**Final Justification:**

The response addresses my concerns, and the paper can meet the acceptance standards. Thus, I keep my scores.

**Limitations:**

yes

**Paper Formatting Concerns:**

Nope

**Quality:**

3

**Strengths And Weaknesses:**

# Strengths
1. The core contribution, "anchor state grouping," is a practical way to get fine-grained credit signals without the usual costs. It's an elegant solution that avoids the prohibitive expense of extra rollouts or auxiliary critic models, which is a big win when training these large-scale agents.
2. A convincing set of experiments. They don't just test on one toy problem; they show consistent, significant gains on tough benchmarks like ALFWorld and WebShop against a strong field of baselines, including GRPO itself. The numbers speak for themselves, with reported boosts of over 12% on ALFWorld and 9% on WebShop over the GRPO baseline, which is quite impressive. I also appreciate the deeper analysis provided; the ablation studies effectively demonstrate that both the macro and micro advantage components are necessary for the method's success, and the plot showing how group sizes evolve during training was a nice touch that gives real insight into the learning dynamics.
3. the paper is very well-written.
# Weaknesses
1. My main concern is the robustness of the anchor state grouping. The current method relies on exact state matching, which seems to work well enough in the clean, simulated environments tested6. But how would this hold up in the wild? In messier, noisier settings with minor but constant state variations (e.g., dynamic content on a webpage), this hashing approach would likely fail often, causing the method to simply fall back to GRPO. The authors do acknowledge this in their conclusion. but the paper would be much more compelling if it included an experiment that actually tests this degradation, perhaps by introducing controlled noise into the state representations. That would give us a much better sense of the method's practical limits.
2. Secondly, the paper is missing a key ablation study for the hyperparameter ω. This coefficient in Equation (8) balances the macro (episode) and micro (step) advantage signals and seems quite important to the whole setup. Just setting it to 1 "with no further tuning" feels like a missed opportunity. It's very likely that the optimal balance is task-dependent, and an analysis of how performance changes with different values of  ω would add a lot of value for anyone looking to apply this method.
3. Finally, the assumption of identical initial states for all trajectories in a group, while standard for this kind of RL, does limit the method's immediate applicability. It's fine for simulations, but it's a real barrier for many real-world scenarios where you can't just reset the world to the exact same starting point. A more explicit discussion on how this limitation might be relaxed in future work would be a welcome addition to the paper.

---

> ### Author Rebuttal · Authors · 2025-07-31
>
> We sincerely appreciate your in-depth and encouraging feedback.
> Thank you for highlighting that our method provides “an elegant solution” for obtaining fine-grained credit signals and finding the experimental results convincing.
> We also appreciate your kind comments on the clarity of the paper and the “deeper analysis” offered through ablation studies and training dynamics visualizations.
> Regarding your concerns, we provide our responses below:
>
> > (Weakness 1 & Question 2) The robustness of the anchor state grouping.
>
> Thank you for this insightful suggestion. As discussed in Sec. 6 (Conclusions and Limitations), we can use **state similarity** to better capture structurally equivalent states in noisy scenarios.
> We have explored this perspective and conducted experiments to evaluate how GiGPO performs under realistic noise, where exact state matches are rare. We find **using similarity for GiGPO works well**.
>
> Specifically:
> - (Noisy WebShop): To simulate realistic noise, we used GPT-4o to generate 551 diverse advertisements, which are **randomly inserted** into the states returned by WebShop to **simulate noise from ad pop-ups**. This substantially increases the state spaces.
> - (Similarity‐based grouping): We use Ratcliff–Obershelp similarity (ranges from 0 to 1) to construct step-level groups: **grouping the states whose similarity** $\geq$ `sim_thresh`. Exact match is the special case `sim_thresh=1.0`.
>
> The results are shown as follows:
>
> ||Score|Success Rate |Avg. Step Group Size|
> |:-:|:-:|:-:|:-:|
> |GiGPO (sim_thresh=1.0)|74.5|56.7|1.01|
> |GiGPO (sim_thresh=0.9)|82.6|66.0|4.50|
> |GiGPO (sim_thresh=0.8)|83.2|66.2|4.96|
> |GiGPO (sim_thresh=0.5)|80.7|64.8|6.69|
> |GiGPO (sim_thresh=0.2)|78.7|58.6|12.15|
>
> We can see that:
> - **Exact-match failure**. With `sim_thresh=1.0`, the avg. step group size collapses to $\approx 1$, indicating that almost no identical states are found. As a result, GiGPO **degrades to GRPO** and performance drops.
> - **GiGPO remains robust and effective when using similarity-based grouping** even under noisy conditions. As shown, `sim_thresh=0.9/0.8` works well in such a setting. As `sim_thresh` decreases further (e.g., 0.5 or 0.2), group size increases, but similarity quality diminishes, revealing a trade-off between group generalization and precision.
>
> > (Weakness 2 & Question 1) Ablation study for the hyperparameter $\omega$
>
> Thanks for your valuable feedback. We conduct the sensitivity study on the hyperparameter $\omega$ in WebShop.
> | $\omega$           | 0.0  | 0.2  | 0.4  | 0.6  | 0.8  | 1.0  | 1.2  | 1.4  |
> |--------------------|------|------|------|------|------|------|------|------|
> | Score              | 76.2 | 79.6 | 82.4 | 83.5 | 84.9 | 83.5 | 82.6 | 77.0 |
> | Success Rate | 56.6 | 63.1 | 65.8 | 67.2 | 68.3 | 67.4 | 66.5 | 56.3 |
>
> We find that
> - **GiGPO needs an appropriate $\omega$ to work best**.
> - Increasing $\omega$ **initially improves** performance due to the added fine-grained step-level reward. However, performance **declines beyond the optimum** ($\omega=0.8$), suggesting that excessive emphasis on step-level signals can suppress useful trajectory-level guidance.
> - GiGPO is **relatively insensitive** to $\omega$ within the range $[0.4, 1.2]$, demonstrating a reasonable degree of robustness to $\omega$.
>
> > (Weakness 3) The assumption of identical initial states
>
> - The assumption of identical beginning is a **standard setting in all group-based RL methods** such as GRPO, RLOO, and DAPO. GiGPO is designed for group-based RL and therefore **follows this established setting**.
> - We also explored the **non-identical** initial state case in our response to (Weakness 1 & Question 2), where we introduced randomized ad pop-ups to perturb the initial states. GiGPO remained effective using similarity-based grouping, suggesting that **exact initial state matching might not be strictly necessary**.
> - This indicates that knowledge sharing is feasible among similar tasks, providing promise for relaxing this assumption in future work.
>
>
> Thank you again for all your constructive feedback. We will include these clarifications, especially the experiments on robustness to noise and the hyperparameter sensitivity analysis.

---

> > ### Comment · Reviewer_yaqn · 2025-08-04
> >
> > Thanks for your reply and it resolves my concerns. I will keep the accept score.

---

> > > ### Author Response · Authors · 2025-08-05
> > > **Thanks to Reviewer yaqn**
> > >
> > > Thank you very much for your positive feedback and constructive suggestions. We are delighted to see that our response has addressed all your concerns and sincerely hope this might encourage a positive update of your initial assessment within the review panel.
> > >
> > > As the discussion period has been extended to 8 Aug, there is still ample time for further exchange. If you still have any questions or reservations about our work, we are more than willing to engage in further discussion with you.

---

### Note · Authors · 2025-08-14

We deeply appreciate all reviewers’ time and effort in reviewing our paper and providing constructive feedback. We are greatly encouraged that all reviewers gave positive scores.

Besides our detailed point-by-point responses, here we would like to 1) further thank the reviewers for their recognition of our work; and 2) summarize the main concerns we have addressed during the rebuttal.

**(1) We are glad that the reviewers appreciate and recognize our key contributions.**
- Novelty, practicality, and simplicity of GiGPO's design for fine-grained credit assignment in multi-turn LLM training [**yaqn, aEAm, fP6w**].
- Accurate problem identification with strong motivation, clearly targeting the credit assignment challenge in multi-step RL [**fP6w**].
- Anchor state grouping as an elegant and efficient way to obtain high-quality advantage signals without extra rollouts or critic models [**yaqn, vKcT, fP6w**].
- Strong empirical results with consistent gains over competitive baselines across challenging benchmarks [**yaqn, vKcT, fP6w**].
- Insightful analyses, including macro–micro ablations and training dynamics visualizations that deepen understanding of GiGPO’s behavior [**yaqn**].
- Clarity and high-quality presentation, with reviewers highlighting the writing quality, clear structure, and effective figures/tables [**yaqn, vKcT, aEAm, fP6w**].


**(2) We have carefully addressed all reviewers' concerns with additional experiments, analyses, and clarifications.**
- Noisy and complex environments. We showed that GiGPO remains robust in more realistic, noisy settings (noisy WebShop with ad pop-ups), and maintains strong performance across thresholds [**yaqn, vKcT, aEAm, fP6w**].
- Sensitivity study on $\omega$. Our results revealed an optimum at $\omega=0.8$ and robustness within $[0.4, 1.2]$ [**yaqn, fP6w**].
- Additional baselines. Added DeepSeek-R1-0528 and RAGEN; GiGPO outperformed both [**fP6w**].
- Short-horizon or open environments. We have demonstrated GiGPO’s effectiveness in Search-R1 experiments, confirming its benefits beyond long-horizon tasks [**vKcT, fP6w**].
- Generality and the applicable tasks. We have clarified that GiGPO can be applied to all group-based RL frameworks and all multi-turn LLM agent tasks [**aEAm**].

Throughout the rebuttal period, we have made our best efforts and shown our utmost sincerity in addressing the reviewers’ concerns. We greatly appreciate the reviewers’ and ACs’ time and thoughtful engagement with our work.

---

### Decision · Program_Chairs · 2025-09-17

**Decision:**

Accept (poster)

**Comment:**

This paper introduces Group-in-Group Policy Optimization (GiGPO), a simple, critic-free RL algorithm for long-horizon LLM agents that combines (i) episode-level relative advantages with (ii) step-level advantages computed by retroactively grouping identical environment states. The combined objective (Eq. 8) provides hierarchical credit signals without extra rollouts or an auxiliary value model, and delivers strong gains on ALFWorld and WebShop across 1.5B and 7B backbones.

**Strengths**
* **Clear problem framing & elegant method.** The paper targets the key gap in group-based RL for multi-turn LLM agents—fine-grained credit assignment—via a two-level advantage that is conceptually simple and implementation-friendly (critic-free, no extra rollouts).
* **Compelling empirical results.** On ALFWorld and WebShop, GiGPO consistently beats strong prompting, PPO, RLOO, and GRPO baselines.
* **Ablations that isolate what matters.** Removing either episode- or step-level advantages notably degrades performance, supporting the necessity of both macro and micro signals.
* **Clarity and presentation quality.** Reviewers uniformly praise the writing and structure; the paper also details environments, baselines, and training settings.

**Weaknesses raised in reviews & how the rebuttal addressed them**
* **Fragility of exact-match anchor grouping in noisy states.** Reviewers worry the hash-match assumption would collapse in real-world noise. The authors run a Noisy WebShop variant and switch to similarity-based grouping. Results show robustness with similarity thresholds 0.8–0.9, improving success vs. exact match. This both diagnoses failure at exact match and offers a practical fix with a precision-vs-group-size trade-off.
* **Sensitivity to the macro/micro weight $\omega$**. The revised analysis on WebShop finds an optimum around $\omega\approx0.8$ and robustness for $\omega\in[0.4,1.2]$, providing concrete guidance for practitioners and showing that performance is not knife-edge sensitive.
* **Applicability beyond long-horizon / identical initial states.** The rebuttal adds short-horizon Search-R1 experiments where GiGPO still outperforms GRPO, showing benefit even with only ~4 steps. The authors also note that similarity-based grouping relaxes strict initial-state identity.
* **Missing/stronger baselines and compute practicality.** The authors add DeepSeek-R1 and RAGEN comparisons; GiGPO remains stronger. They also clarify compute: the H100×2 setup shortens wall-clock but is not required; training can run on a single H100 (+ longer time) with unchanged results.

The paper presents a clear, scalable solution to long-horizon credit assignment for LLM agents with solid evidence and constructive rebuttal additions. As a result, I recommend accept.